

# Measurements of wind turbulence parameters by a conically scanning coherent Doppler lidar in the atmospheric boundary layer

Igor N. Smalikho, Viktor A. Banakh

V.E. Zuev Institute of Atmospheric Optics SB RAS, Tomsk, Russia

*Correspondence to*: Igor N. Smalikho (smalikho@iao.ru)

**Abstract.** The method and results of lidar studies of spatiotemporal variability of wind turbulence in the atmospheric boundary layer are reported. The measurements were conducted by a Stream Line pulsed coherent Doppler lidar with the use of conical scanning by a probing beam around the vertical axis. Lidar data are used to estimate the kinetic energy of turbulence, turbulent energy dissipation rate, integral scale of turbulence, and momentum fluxes. The dissipation rate was

determined from the azimuth structure function of radial velocity within the inertial subrange of turbulence. When estimating the kinetic energy of turbulence from lidar data, we took into account the averaging of radial velocity over the sensing volume. The integral scale of turbulence was determined on the assumption that the structure of random irregularities of the wind field is described by the von Karman model. The domain of applicability of the used method and the accuracy of estimation of turbulence parameters were determined. Turbulence parameters estimated from Stream Line lidar measurement

data and from data of a sonic anemometer were compared.

## 1 Introduction

Pulsed coherent Doppler lidars (PCDLs) are applied in various fields of scientific research, in particular, to study dynamic processes in the atmosphere, aircraft wake vortices, and wind turbine wakes (Banakh and Smalikho, 2013). PCDLs are quite promising for obtaining reliable estimates of wind turbulence parameters from raw lidar data measured in the entire

atmospheric boundary layer (Eberhard et al., 1989; Gal-Chen and Eberhard, 1992; Frehlich et al., 1998; Frehlich and Cornman, 2002; Davies et al., 2004; Smalikho et al., 2005; Banta et al., 2006; Frehlich et al., 2006; O'Connor et al., 2010; Banakh and Smalikho, 2013; Sathe and Mann, 2013; Smalikho and Banakh, 2013; Smalikho et al., 2013; Sathe et al., 2015). For this purpose, different measurement geometries were proposed, and methods were developed for estimation of turbulence parameters, in particular, with allowance made for averaging of the radial velocity over the sensing volume

and for the instrumental measurement error. Here, the radial velocity $V_r$ is understood as a projection of the wind vector $\mathbf{V} = \{V_z, V_x, V_y\}$ ($V_z$ is the vertical component, $V_x$ and $V_y$ are the horizontal components) onto the axis of the probing beam at the point $\mathbf{r} = \{z, x, y\} = R\mathbf{S}$, where $R$ is the distance from the lidar, $\mathbf{S} = \{\sin\varphi, \cos\varphi\cos\theta, \cos\varphi\sin\theta\}$, $\varphi$ is the elevation




angle, and $\theta$ is the azimuth angle. Denote the average wind velocity and the wind direction angle as $U$ and $\theta_V$, respectively, and fluctuations of the vertical, longitudinal, and transverse wind components as $w$, $u$, and $v$.

The use of the conical scanning by the probing beam (when the elevation angle $\varphi$ is fixed during measurements, while the azimuth angle $\theta = \omega_s t$ varies in time $t$ with the constant angular rate $\omega_s$) allows reconstruction of not only the wind speed
and direction, but also vertical profiles of different wind turbulence parameters from raw data measured by PCDL. It was shown by Eberhard et al. (1989) that the kinetic energy of turbulence $E = (\sigma_w^2 + \sigma_u^2 + \sigma_v^2)/2$ can be determined from measurements by conically scanning PCDL at the elevation angle $\varphi = 35.3°$, where $\sigma_w^2 = <w^2>$, $\sigma_u^2 = <u^2>$, $\sigma_v^2 = <v^2>$, and the angular brackets denote the ensemble averaging. However, in the results for $E$, the effect of averaging of the radial velocity over the sensing volume was not taken into account. A method for reconstructing the vertical profiles
of the fluxes of momentum $<uw>$ and $<vw>$ was also proposed by Eberhard et al. (1989).

Methods for determination of the turbulent energy dissipation rate $\varepsilon$ and the integral scale of turbulence $L_V = \int_0^\infty dr B_\parallel(r)/\sigma_r^2$, where $B_\parallel(r)$ is the longitudinal correlation function of wind velocity, from measurements by conically scanning PCDL were proposed (Frehlich et al., 2006; Smalikho and Banakh, 2013; Smalikho et al. 2013). In this case, turbulence parameters are estimated through fitting of the theoretically calculated azimuth (transverse) structure function of
the radial velocity measured by the lidar to the corresponding measured function on the assumption that turbulence is isotropic and its spatial structure is described by the von Karman model (Vinnichenko et al., 1973). However, if the radius of the scanning cone base $R' = R\cos\varphi$, where $R$ is the distance between the lidar and the center of the sensing volume, is comparable with or smaller than $L_V$, then the method of the azimuth structure function can give a large error in estimates of wind turbulence parameters (Smalikho and Banakh, 2013).
Pulsed coherent Doppler lidars capable of providing measured data with high spatial resolution, for example, Stream Line lidars (HALO Photonics) and Windcube lidars (Leosphere) are now widely used in practice. In this paper, for lidars of this type, we propose a method for determination of wind turbulence parameters from measurements by conically scanning PCDLs, which removes the mentioned disadvantages of the earlier methods. With the use of the proposed method, we have obtained the spatiotemporal distributions of $E$, $\varepsilon$, $L_V$, $<uw>$, and $<vw>$ in the atmospheric layer from 100 to 500 m
from data of an atmospheric experiment with the Stream Line lidar. The accuracy of the obtained results is analyzed.

## 2 Basic equations

First of all, derive the equations to be used as a basis for development of the measurement strategy and the procedure of estimation of wind turbulence parameters: $E$, $\varepsilon$, and $L_V$. Instantaneous values of components of the wind velocity vector are random functions of coordinates and time, that is, $\mathbf{V} = \mathbf{V}(\mathbf{r}, t)$. The radial velocity at a point moving in the cone base of



conical scanning as the azimuth angle $\theta$ changes from 0° to 360° (or in radians from 0 to $2\pi$) can be represented in the form

$$V_r(\theta) = \mathbf{S}(\theta) \cdot \mathbf{V}(R\mathbf{S}(\theta), \theta / \omega_s),\tag{1}$$

where $\varphi$, $R$, and $\omega_s$ are constant parameters.

The turbulence is assumed to be stationary (for time scales no shorter than 1 hour) and horizontally homogeneous (within the scanning cone base). Because of anisotropy of wind turbulence, the variance of the radial velocity $\sigma_r^2 = <[V_r'(\theta)]^2 >$, where $V_r' = V_r - <V_r>$, is a function of the azimuth angle: $\sigma_r^2 = \sigma_r^2(\theta)$. For the variance of the radial velocity averaged over the azimuth angles

$$\bar{\sigma}_r^2 = (2\pi)^{-1} \int_0^{2\pi} d\theta\, \sigma_r^2(\theta),\tag{2}$$

from Eqs. (1) and (2) after the corresponding ensemble averaging and integration over the angle $\theta$, we obtain the equation

$$\bar{\sigma}_r^2 = (\sin\varphi)^2 \sigma_w^2 + (1/2)(\cos\varphi)^2 (\sigma_u^2 + \sigma_v^2).\tag{3}$$

From Eq. (3) at the angle $\varphi = \varphi_E = \tan^{-1}(1/\sqrt{2}) \approx 35.3°$, we can find a simple relation between the kinetic energy of turbulence $E$ and the variance $\bar{\sigma}_r^2$ in the form (Eberhard et al., 1989)

$$E = (3/2)\bar{\sigma}_r^2.\tag{4}$$

Consider the azimuth structure function of the radial velocity $D_r(\psi;\theta) = <[V_r'(\theta+\psi) - V_r'(\theta)]^2 >$ ( $\psi > 0$ ). For this function at $\psi \leq \pi/2$ (90°) and the fast movement of a point in a circle of the radius $R' = R\cos\varphi$, when the condition $R'\omega_s >> |<\mathbf{V}>|$ is true, the transfer of turbulent inhomogeneities by the average flow can be neglected. Due to anisotropy of turbulence, the function $D_r(\psi;\theta)$, in the general case, depends on the angle $\theta$. By analogy with Eq. (2), we introduce the averaged structure function

$$\bar{D}_r(\psi) = (2\pi - \psi)^{-1} \int_0^{2\pi-\psi} d\theta\, D_r(\psi;\theta).\tag{5}$$

Under the condition $\psi R' << L_V$, due to the local isotropy of turbulence, $D_r(\psi;\theta)$ is independent of $\theta$, and $\bar{D}_r(\psi) = D_r(\psi)$. In addition, if the condition $R' > L_V$ is also fulfilled, then, according to the Kolmogorov theory, $D_r(\psi)$ is described by the equation (Kolmogorov, 1941)

$$D_r(\psi) = (4/3)C_K(\varepsilon\psi R')^{2/3},\tag{6}$$





where $C_K \approx 2$ is the Kolmogorov constant.

To find the relation between the structure function $\bar{D}_r(\psi)$ and the integral scale $L_V$, it is necessary to know the equation for the correlation tensor of wind turbulence $B_{\alpha\beta}(\mathbf{r}) = <V'_\alpha(\mathbf{r}_0 + \mathbf{r})V'_\beta(\mathbf{r}_0)>$ ($\alpha, \beta = z, x, y$; $V' = V - <V>$), which can be readily found for the case of isotropic turbulence using some or other model for $B_\parallel(r)$. To find this relation, we assume that

turbulence is isotropic, and within this assumption $\bar{D}_r(\psi) = D_r(\psi)$. Upon generalization of Eq. (19) given in (Smalikho and Banakh, 2013) for $\varphi = 0°$, for the case of an arbitrary elevation angle, we have derived the following equation

$$D_r(\psi) = 4\int_0^\infty d\kappa S_\parallel(\kappa)[1 - \mu_1 \cos(2\pi r'\kappa) + \mu_2 \pi r'\kappa \sin(2\pi r'\kappa)], \tag{7}$$

where     $\mu_1 = (\cos\varphi)^2 \cos\psi + (\sin\varphi)^2$     ,     $\mu_2 = (\cos\varphi)^2(1 + \cos\psi)/2 + (\sin\varphi)^2$     ,     $r' = R'\sqrt{2(1 - \cos\psi)}$     ,     and

$S_\parallel(\kappa) = 2\int_0^\infty dr B_\parallel(r)\cos(2\pi\kappa r)$ is the longitudinal spatial spectrum of wind velocity fluctuations. If the condition

$R' = R\cos\varphi \gg L_V$ is fulfilled, in Eq. (7) we can set $\mu_1 = \mu_2 = 1$, $r' = y' = \psi R'$ (here, the angle $\psi$ is in radians), and then for any angles $\psi \leq 180°$ the azimuth structure function $D_r(\psi)$ coincides with the transverse structure function

$$D_\perp(y') = 4\int_0^\infty d\kappa S_\perp(\kappa)[1 - \cos(2\pi y'\kappa)], \tag{8}$$

where $S_\perp(\kappa) = [S_\parallel(\kappa) - \kappa dS_\parallel(\kappa)/d\kappa]/2$ is the transverse spectrum of wind velocity fluctuations (Lumley and Panofsky, 1964; Monin and Yaglom, 1971).

For the spectrum $S_\parallel(\kappa)$, we use the von Karman model (Vinnichenko et al., 1973; Smalikho and Banakh, 2013):

$$2\sigma_r^2 L_V[1 + (C_1 L_V \kappa)^2]^{-5/6}, \tag{9}$$

where $C_1 = 8.4134$. For this model, the following relationship is true

$$\sigma_r^2 = C_2(\varepsilon L_V)^{2/3}. \tag{10}$$

In Eq. (10) at $C_K = 2$, the coefficient $C_2 = 1.2717$ (Smalikho and Banakh, 2013).

Figure 1 shows the results of calculation of the normalized structure functions $D_r(\psi)/\sigma_r^2$ and $D_\perp(R'\psi)/\sigma_r^2$ at $\varphi = \varphi_E \approx 35.26°$ and different values of the ratio $R'/L_V$. It can be seen that the higher the ratio, the smaller the difference between the functions. Calculations at $R'/L_V \geq 4$ demonstrate the nearly complete coincidence of the structure functions described by Eqs. (7) and (8) for any angles $\psi \leq 180°$. The nearly complete coincidence is also observed at $\psi \leq 9°$ for any



$R'/L_V \geq 1/4$. At the same time, if the condition $R'/L_V << \psi^{-1}$ is fulfilled, then, with allowance for Eq. (10), the both structure functions $D_r(\psi)$ and $D_\perp(R'\psi)$ are described by Eq. (6).

We introduce the parameter $\gamma$ characterizing the degree of deviation of $D_\perp(R'\psi)$ from $D_r(\psi)$ as

$$\gamma = \{L^{-1}\sum_{l=1}^{L}[D_r(l\Delta\theta)/D_\perp(R'l\Delta\theta)-1]^2\}^{\frac{1}{2}}, \tag{11}$$

where $\Delta\theta = 3°$ and $L = 30$. Using the data of Fig. 1, we have calculated the parameter $\gamma$ by this equation and obtained the following results: $\gamma = 0.21$ at $R'/L_V = 0.5$; $\gamma = 0.08$ at $R'/L_V = 1$, and $\gamma = 0.02$ at $R'/L_V = 2$. It should be noted that if we fit the function $D_\perp(R'l\Delta\theta)$ with arbitrary values of $\varepsilon$ and $L_V$ by the least-square method (see Eqs. (13)–(16) in paper of Smalikho and Banakh (2013)) to the function $D_r(l\Delta\theta)$ obtained at $R'/L_V = 0.5$, then we can attain a significant decrease in the parameter $\gamma$ (six times in comparison with the above values), but the estimates of $L_V$ and $\sigma_r^2$ exceed the true values of

these parameters more than twice, although the error of $\varepsilon$ estimation by this method is about 15%. Therefore, for these situations (when the ratio $R'/L_V < 1$), it is possible to obtain the more accurate result through direct determination of the variance $\sigma_r^2$ and the dissipation rate $\varepsilon$ (the dissipation rate is determined from the azimuth structure function of the radial velocity within the inertial subrange of turbulence with the use of Eq. (6)) and then calculation of the integral scale $L_V$ by Eq. (10).

**3 Measurement strategy and estimation of turbulence parameters**

To obtain the information about the kinetic energy, its dissipation rate, and the integral scale of turbulence from the same raw lidar data, it is proposed, according to the previous section, to use the conical scanning by the probing beam at the elevation angle $\varphi = \varphi_E \approx 35.3°$ in the experiment. During the measurements, the azimuth angle changes starting from 0° with the constant angular rate $\omega_s = 2\pi/T_{scan}$, where $T_{scan}$ is the time of one scan. As an angle of 360° is achieved, the scanning in

the opposite direction starts practically immediately. This cycle is repeated many times during the experiment.

An array of estimates of the radial velocity $V_L(\theta_m, R_k, n)$ is obtained from signals recorded by the PCDL receiving system after the corresponding pre-processing (Banakh et al., 2016). Here, $\theta_m = m\Delta\theta$ is the azimuth angle; $m = 0, 1, 2, ..., M-1$; $\Delta\theta$ is the azimuth resolution; $R_k = R_0 + k\Delta R$ is the distance from the lidar to the center of the sensing volume; $k = 0, 1, 2, ..., K$; $\Delta R$ is a range gate length, and $n = 1, 2, 3, ..., N$ is the number of full conical scans. The minimal distance

$R_0$ depends on the probing pulse duration. At the same time, it should satisfy the above condition $R_0 >> |< \mathbf{V} >|/(\omega_s \cos\varphi_E)$. The maximal distance $R_K$ is determined by the lidar signal-to-noise ratio $SNR$, at which the probability of "bad" estimate of the radial velocity randomly taking any values in the chosen receiver band (for example, $\pm 19,4$ m/s for the Stream Line





lidar) is practically zero regardless of the true value of the velocity. Then, the lidar estimate of the radial velocity can be represented as (Frehlich and Cornman, 2002; Banakh and Smalikho, 2013)

$$V_L(\theta_m, R_k, n) = V_a(\theta_m, R_k, n) + V_e(\theta_m, R_k, n) \,, \tag{12}$$

where $V_a(\theta_m)$ is the radial velocity averaged over the sensing volume with the longitudinal dimension $\Delta z$ and the transverse

dimension $\Delta y_k = \Delta \theta R_k \cos \varphi_E$ (here, $\Delta \theta$ is in radians), and $V_e(\theta_m)$ is the random instrumental error of estimation of the radial velocity having the following properties: $<V_e> = <V_a V_e> = 0$ and $<V_e(\theta_m)V_e(\theta_l)> = \sigma_e^2 \delta_{m-l}$ ($\sigma_e^2$ is the variance of random error, $\delta_m$ is the Kronecker delta). For the conditions of stationary and homogeneous turbulence, the estimate is unbiased, that is, $<V_L(\theta_m, R_k, n)> = <V_r(\theta_m, R_k)>$.

Lidars of Stream Line type, one of which is used in our experiments, are characterized by formation of a sensing volume

of relatively small size, for example, with the longitudinal dimension $\Delta z = 30$ m (Pierson et al., 2009). When the conical scanning with $\varphi = \varphi_E$ and $\Delta \theta = 3°$ is used, the longitudinal dimension of the sensing volume increases linearly from 8.5 m at $R_k = 200$ m to 42.8 m at $R_k = 1$ km. It is important to take into account the effect from averaging of the radial velocity over the sensing volume not only when estimating the dissipation rate $\varepsilon$ within the inertial subrange of turbulence, but also when estimating the parameters $E$ and $L_V$, especially, when $L_V$ only few times exceeds the size of the sensing volume.

Even at the high signal-to-noise ratio and the large number of probing pulses used for accumulation of lidar data, when the variance $\sigma_e^2$ is extremely small, it is necessary to take into account the instrumental error of estimation of the radial velocity, if turbulence is very weak.

After the corresponding manipulations, from Eq. (12) we can derive the following equations for the variance and the structure function of lidar estimate of the radial velocity averaged over all azimuth angles:

$$\bar{\sigma}_L^2 = \bar{\sigma}_r^2 - \bar{\sigma}_t^2 + \sigma_e^2 \,, \tag{13}$$

$$\bar{D}_L(\psi_l) = \bar{D}_a(\psi_l) + 2\sigma_e^2 \,, \tag{14}$$

where $\bar{\sigma}_\alpha^2 = M^{-1} \sum_{m=0}^{M-1} \sigma_\alpha^2(\theta_m)$ ; $\bar{D}_a(\psi_l) = (M-l)^{-1} \sum_{m=0}^{M-1-l} D_a(\psi_l, \theta_m)$ subscripts $\alpha = L, r, t, a$ ; $\sigma_\alpha^2(\theta_m) = <[V_\alpha'(\theta_m)]^2>$ ;

$V_\alpha' = V_\alpha - <V_r>$ ; $\sigma_t^2(\theta_m) = \sigma_r^2(\theta_m) - \sigma_a^2(\theta_m)$ is turbulent broadening of the Doppler spectrum (Banakh and Smalikho, 2013); $D_\alpha(\psi_l, \theta_m) = <[V_\alpha'(\theta_m + \psi_l) - V_\alpha'(\theta_m)]^2>$ ; $\psi_l = l\Delta \theta$ and $l = 1, 2, ..., L$.

Having specified the high resolution in the azimuth angle (large number $M$) and $\varphi = \varphi_E$, from Eqs. (13) and (14) with allowance made for Eq. (4), we obtain the equation for the kinetic energy of turbulence in the form

$$E = (3/2)[\bar{\sigma}_L^2 - \bar{D}_L(\psi_1)/2 + G] \,, \tag{15}$$





where $G = \bar{\sigma}_t^2 + \bar{D}_a(\psi_1)/2$. At $L_V > \max\{\Delta z, \Delta y_k\}$, the dimensions of the sensing volume do not exceed the low-frequency boundary of the inertial subrange, for which turbulence is locally isotropic and, correspondingly, $G \sim \varepsilon^{2/3}$. If the condition $l\Delta y_k \ll L_V$ is additionally fulfilled, then for calculation of the turbulent broadening of the Doppler spectrum $\bar{\sigma}_t^2 = \sigma_t^2$ and the structure function $\bar{D}_a(\psi_l) = D_a(\psi_l)$ we can use the two-dimensional spatial Kolmogorov—Obukhov spectrum. For these conditions, the Gaussian temporal profile of the probing pulse, and the rectangular time window used for obtaining of Doppler spectra, we have derived the following equations:

$$\sigma_t^2 = \varepsilon^{2/3} F(\Delta y_k), \tag{16}$$

$$D_a(\psi_l) = \varepsilon^{2/3} A(l\Delta y_k). \tag{17}$$

In Eqs. (16) and (17)

$$F(\Delta y_k) = \int_0^\infty d\kappa_1 \int_0^\infty d\kappa_2 \Phi(\kappa_1, \kappa_2)[1 - H_\parallel(\kappa_1)H_\perp(\kappa_2)], \tag{18}$$

$$A(l\Delta y_k) = 2\int_0^\infty d\kappa_1 \int_0^\infty d\kappa_2 \Phi(\kappa_1, \kappa_2)H_\parallel(\kappa_1)H_\perp(\kappa_2)[1 - \cos(2\pi l\Delta y_k\kappa_2)], \tag{19}$$

where $\Phi(\kappa_1, \kappa_2) = C_3(\kappa_1^2 + \kappa_2^2)^{-4/3}[1 + (8/3)\kappa_2^2 / (\kappa_1^2 + \kappa_2^2)]$; $C_3 = 4C_2 / (6\pi C_1^{2/3})$ = 0.0652; $H_\parallel(\kappa_1) = [\exp\{-(\pi\Delta p\kappa_1)^2\}\mathrm{sinc}(\pi\Delta R\kappa_1)]^2$ is the longitudinal transfer function of the low-frequency filter, and $H_\perp(\kappa_2) = [\mathrm{sinc}(\pi\Delta y_k\kappa_2)]^2$ is the transverse one; $\Delta p = c\sigma_p / 2$; $c$ is the speed of light; $2\sigma_p$ is the duration of the probing pulse determined by the $e^{-1}$ power level to right and to the left from the peak point, $\Delta R = cT_W / 2$, $T_W$ is the temporal window width; and $\mathrm{sinc}(x) = \sin x / x$.

In Eq. (15), $\bar{\sigma}_L^2$ and $\bar{D}_L(\psi_1)$ are directly determined from experimental data. To take into account the term $G = \varepsilon^{2/3}[F(\Delta y_k) + A(\Delta y_k)/2]$ in Eq. (15), it is necessary to have information about the dissipation rate $\varepsilon$. According to Eq. (14), the difference $\bar{D}_L(\psi_l) - \bar{D}_L(\psi_1)$ is equal to the difference $\bar{D}_a(\psi_l) - \bar{D}_a(\psi_1)$. Within the framework of the above conditions and according to Eq. (17), the latter is equal to $\varepsilon^{2/3}[A(l\Delta y_k) - A(\Delta y_k)]$. Then the dissipation rate can be determined as

$$\varepsilon = \left[\frac{\bar{D}_L(\psi_l) - \bar{D}_L(\psi_1)}{A(l\Delta y_k) - A(\Delta y_k)}\right]^{3/2}, \tag{20}$$

where the number $l > 1$ should be so that, on the one hand, the consideration is within the inertial subrange and, on the other hand, the condition



$$[\bar{D}_L(\psi_l) - \bar{D}_L(\psi_1)] \gg \bar{D}_L(\psi_1)\sqrt{2/(MN)} \tag{21}$$

is fulfilled. This condition provides for the high accuracy of estimation of the dissipation rate at the large numbers $M$ and $N$. In parallel, we can calculate the instrumental error of estimation of the radial velocity $\sigma_e$ as

$$\sigma_e = \sqrt{[\bar{D}_L(\psi_1) - \varepsilon^{2/3}A(\Delta y_k)]/2} \equiv \sqrt{\frac{\bar{D}_L(\psi_1)A(l\Delta y_k) - \bar{D}_L(\psi_l)A(\Delta y_k)}{2[A(l\Delta y_k) - A(\Delta y_k)]}}. \tag{22}$$

Using the lidar estimates of the kinetic energy $E$ (by Eq. (15)) and the dissipation rate $\varepsilon$ from experimental data, we can determine the integral scale $L_V$ by Eqs. (4) and (10) as

$$L_V = C_4 E^{3/2}/\varepsilon, \tag{23}$$

where $C_4 = [2/(3C_2)]^{3/2} = 0.38$.

Taking into account that the elevation angle $\varphi = \varphi_E = \tan^{-1}\left(1/\sqrt{2}\right)$, we use the following equation (Eberhard et al., 1989)

for determination of the momentum fluxes $<uw>$ and $<vw>$:

$$<uw> + j<vw> = \frac{3}{\sqrt{2}}\frac{1}{M}\sum_{m=0}^{M-1}\sigma_L^2(\theta_m)\exp[j(\theta_m - \theta_V)], \tag{24}$$

where $j = \sqrt{-1}$. Since the instrumental error of estimation of the radial velocity $\sigma_e$ is independent of the azimuth angle $\theta_m$ and within the sensing volume, turbulence is locally isotropic (the condition $L_V > \max\{\Delta z, \Delta y_k\}$ is assumed to be true), it is not necessary here to take into account the instrumental error and the effect from averaging of the radial velocity over the

sensing volume.

The practical implementation of the described method of estimation of the wind turbulence parameters $\varepsilon$, $E$, $L_V$, $<uw>$, and $<vw>$ consists in the following. The obtained array $V_L(\theta_m, R_k, n)$ for every height $h_k = R_k \sin\varphi_E$ was used to determine the average wind vector $<\mathbf{V}>$ (average wind velocity $U$ and wind direction angle $\theta_V$) with the use of the least-square sine-wave fitting and the data of all $N$ scans. Then fluctuations of the radial velocity are calculated as

$V_L'(\theta_m, R_k, n) = V_L(\theta_m, R_k, n) - \mathbf{S}(\theta_m)\cdot<\mathbf{V}>$, where $\mathbf{S}(\theta_m) = \{\sin\varphi_E, \cos\varphi_E\cos\theta_m, \cos\varphi_E\sin\theta_m\}$ (in place of the array $\mathbf{S}(\theta_m)\cdot<\mathbf{V}>$, it is also possible to use directly the calculated values of $V_L'(\theta_m, R_k, n) = V_L(\theta_m, R_k, n) - <V_L(\theta_m, R_k, n)>$ at the nonideal horizontal homogeneity of the average wind). Here and in Eqs. (13)-(15), (20)-(22), and (24), the ensemble averaging $<X>$ should be replaced with the averaging over scans $N^{-1}\sum_{n=1}^{N}X_n$. The number of scans $N$ necessary for the averaging of data was determined experimentally (see Section 5).



To test the described method for measurement of the wind turbulence parameters, we have conducted experiments with the conically scanning Stream Line lidar (the main parameters of the lidar can be found in Table 1 of paper of Banakh and Smalikho (2016)) and the sonic anemometer at a height of 43 m in 2014 and 2016.

**4 Experiment of 2014**

To study the feasibility of estimating the turbulence energy dissipation rate from PCDL data by the method described in Section 3 under various atmospheric conditions, we have conducted the five-day experiment in August 15-19 of 2014 at the Basic Experimental Complex (BEC) of Institute of Atmospheric Optics SB RAS. Experimental instrumentation included the Stream Line PCDL set at the central part of BEC mostly surrounded by forest and a sonic anemometer installed at the top of a tower (near BEC) at a height of 43 m from the ground. The separation between the lidar and the tower was 142 m (see Fig.2).

Conical scanning by the probing beam with an angular rate of 5°/s (duration of one scan $T_{scan} = 72$ s) at the elevation angle $\varphi = 9°$ was applied permanently during the experiment. For accumulation, $N_a = 3000$ of probing pulses were used. Since the pulse repetition frequency of the Stream Line lidar is $f_p = 15$ kHz, the measurement for every azimuth scanning angle took $N_a / f_p = 0.2$ s. In this case, for one scan we have $M = T_{scan} / (N_a / f_p) = 360$ of such measurements with the resolution in the azimuth angle $\Delta\theta = 1°$. Since the lidar telescope is at a height of 1 m above the surface and the elevation angle is 9°,

the probing pulse reaches the height of the sonic anemometer (43 m) at a distance of 270 m. To increase the lidar signal-to-noise ratio in the height of 43 m, we focused the probing radiation to a distance of 300 m. In Fig. 2, the blue circle shows the trajectory of the center of the sensing volume at a height of 43 m during the measurements.

From the array of radial velocities measured by the lidar in four full cycles of conical scanning ( $N = 4$ ) for approximately 5 min (for this time at $R = 270$ m and $\varphi = 9°$, the sensing volume passes the distance $8\pi R \cos\varphi$ equal to about 6.7 km), we

have calculated the values of the azimuth structure function $\bar{D}_L(\psi_1)$ and $\bar{D}_L(\psi_l)$ . We obtained lidar estimates of the turbulent energy dissipation rate $\varepsilon_L$ by Eq. (20) ( $\varepsilon$ should be replaced with $\varepsilon_L$ ). To calculate the longitudinal structure functions $D_\parallel(r_1)$ and $D_\parallel(r_2)$ at separations of observation points $r_1 = \Delta t_1 U$ and $r_2 = \Delta t_2 U$ ( $r_1, r_2 > 0$ , $r_2 > r_1$ ; $\Delta t_1$ and $\Delta t_2$ are time separations), we used the array of longitudinal components of the wind vector measured by the sonic anemometer for the time $T = 20$ min (at a sampling frequency of 10 Hz). For this time, at the average wind velocity $U = 5$ m/s typical

of the surface layer, air masses move to a distance $UT = 6$ km, which is quite comparable with the corresponding value for the lidar data (about 6.7 km). We obtained estimates of the dissipation rate from the sonic anemometer data $\varepsilon_S$ by the equation

$$\varepsilon_S = \left[ \frac{D_\parallel(r_2) - D_\parallel(r_1)}{C_K \cdot (r_2^{2/3} - r_1^{2/3})} \right]^{3/2} ,$$

(25)





on the assumption that $l_v << r_1 < r_2 \leq r_H$, where $l_v$ is the inner scale of turbulence and $r_H$ is the scale of the low-frequency boundary of the inertial subrange. Thus, the sample sizes for the lidar data and the sonic anemometer data are close, and the comparison of estimates of the dissipation rate $\varepsilon_S$ and $\varepsilon_L$ at properly specified $l$, $r_1$, $r_2$ and temporal synchronization of the results is quite justified.

According to the experimental data given in (Byzova et al., 1989), the upper boundary of the inertial subrange $r_H$ at a height of 43 m takes values no smaller than 20 m, at least, at the neutral, unstable, and weak stable temperature stratification of the atmospheric boundary layer. In our case, $\Delta y_k = 4.84$ m, and for $l = 4$ the condition $l\Delta y \leq 20$ m is true. In processing of the sonic anemometer data, we specified $r_1 = 5$ m and $r_2 = 20$ m.

Lidar measurements were started at 18:00 LT on 8/15/2014 and finished at 14:30 LT on 8/19/2014. Unfortunately, because

of the weather conditions (low SNR) and some technical troubles, a part of raw lidar data appeared to be unusable for the processing. Nevertheless, we succeeded in obtaining results under different atmospheric conditions for five days.

All the results of estimation of the turbulent energy dissipation rate from the data measured by the sonic anemometer and the Stream Line lidar are shown in Fig. 3. One can see, in general, a rather good agreement between the results obtained from measurements by these devices. For calculation of the relative errors of estimation of the dissipation rate

$E_S = \sqrt{<(\varepsilon_S / <\varepsilon_S > -1)^2 >} \times 100\%$ and $E_L = \sqrt{<(\varepsilon_L / <\varepsilon_L > -1)^2 >} \times 100\%$, we used the data of Fig. 3 obtained from measurements under relatively steady conditions from 12 to 18 LT on August 18. The errors appeared to be rather close: $E_S = 19\%$ and $E_L = 20\%$.

Using the data of Fig. 3, we have compared all estimates of the turbulent energy dissipation rate obtained from joint (simultaneous) measurements by the lidar and the sonic anemometer. The result of comparison is shown in Fig. 4.

Calculations of parameters characterizing discrepancies in the estimates of the dissipation rate $b_{LS} = <(\varepsilon_L - \varepsilon_S) / [(\varepsilon_L + \varepsilon_S) / 2] > \times 100\%$ and $\Delta_{LS} = \sqrt{<(\varepsilon_L - \varepsilon_S)^2 / [(\varepsilon_L + \varepsilon_S)^2 / 4] >} \times 100\%$ with the use of all points in Fig. 4 have shown that $b_{LS} = -10\%$ and $\Delta_{LS} = 45\%$. Thus, the lidar estimate $\varepsilon_L$ is, on average, 10% smaller than the estimate of the dissipation rate from the data of sonic anemometer. If we assume that random errors of estimates from data of these devices are statistically independent and the variances of random errors are identical, the root-mean-square error of

estimate of the dissipation rate is about 30%, which is 1.5 times higher than the value of $E_L$ given above.

It can be easily seen from Fig. 4 that at $\varepsilon < 10^{-3}$ m$^2$/s$^3$, the lidar estimates of the dissipation rate $\varepsilon_L$ are, on average, understated in comparison with the estimates $\varepsilon_S$. According to Fig.3, the estimates of the dissipation rate taking values smaller than $10^{-3}$ m$^2$/s$^3$ were mostly obtained from nighttime measurements. As a rule, the temperature stratification is stable in nighttime, and then the upper boundary of the inertial subrange $r_H$ can be smaller than the spread in observation points

$4\Delta y_k, r_2 \sim 20$ m taken in Eqs. (20) and (25). In this case, estimates of the dissipation rate from the lidar and sonic



anemometer data are understated, but the lidar estimate is understated to a greater extent because of the averaging of radial velocity over the sensing volume. Using the points of Fig. 4, whose coordinates satisfy the conditions $\varepsilon_S \geq 10^{-3}$ m$^2$/s$^3$ and $\varepsilon_L \geq 10^{-3}$ m$^2$/s$^3$, we have obtained $b_{LS} = 0$ and $\Delta_{LS} = 30\%$. On the assumption of independent estimates from the data of lidar and sonic anemometer and of the equal variances of estimates, the error of lidar estimate of the dissipation rate, which can be calculated as $E_L = \Delta_{LS} / \sqrt{2}$, is equal to 21%. Thus, for the conditions of moderate and strong turbulence, when $\varepsilon \geq 10^{-3}$ m$^2$/s$^3$, the lidar estimate of the turbulence energy dissipation rate is unbiased, while the relative standard error of the estimation is about 20%.

## 5 Experiment of 2016

To test of method for determination of the kinetic energy, its dissipation rate, the integral scale of turbulence, and momentum fluxes as described in Section 3, we have carried out the five-day experiment from 19:00 (from here on, the local time is used everywhere) of July 20 to 15:00 of July 24, 2016, at BEC. The Stream Line lidar was set exactly at the same place as in Experiment of 2014 (see Fig. 2). The weather was clear during these days. The presence of forest fires in the Tomsk region provided for the high concentration of aerosol particles in the atmosphere, which favored the lidar measurements at the rather high signal-to-noise ratio.

The Stream Line lidar operated permanently during the experiment. The probing radiation was focused at a distance of 500 m. The conical scanning with an angular rate of 6°/s (time of one full scan $T_{scan}$ = 1 min) at the elevation angle $\varphi = \varphi_E = 35.3°$ was used. The number of probing pulses for data accumulation was $N_a$ = 7500, which corresponded to the duration of measurement for every azimuth scanning angle $T_a$ = 0.5 s. In this case, for one full scan we have $M = T_{scan} / T_a$ = 120 such measurements with the resolution in the azimuth angle $\Delta\theta$ = 3°. The range gate length $\Delta R$ was taken equal to 18m (vertical resolution $\Delta h = \Delta R \sin\varphi_E \approx 10$ m).

In the processing of raw data of these measurements, we set the minimal distance from the lidar $R_0$ = 171 m, which corresponded to a minimal height of approximately 100 m. Except for the period from 5:00 to 9:00 LT of 7/21/2016, the probability of "bad" lidar estimates of the radial velocity was zero for the ranges from $R_0$ to almost 900 m. The maximum range $R_K$ was taken equal to 873 m, which corresponded to a height of about 500 m. In this experiment, the linear velocity of horizontal motion of the sensing volume (along the base of the scanning cone) $V_k = 2\pi \cos\varphi_E R_k / T_{scan}$ was 14.6 m/s for $R_k = R_0$ and 74.6 m/s for $R_k = R_K$. In this case, for one minute the center of the sensing volume passed a distance of, respectively, 876 m and 4476 m. In Fig. 2, red circles 1 and 2 show the trajectories of the lidar sensing volume at heights of, respectively, 100 and 500 m.

To obtain estimates of the wind turbulence parameters, raw data measured by some or other device for the time of 10 and 60 minutes are usually used. In our case, $T_{scan}$ = 1 min. This corresponds to the use of lidar data obtained for the number of





full conical scans $N$ from 10 to 60. To determine the optimal number $N$, we selected the lidar data measured at night and day on July 22 of 2016 at a height of 200 m (1) from 01:00 to 07:00 and (2) from 12:00 to 18:00 LT. In these six-hour intervals, the wind velocity averaged for 30 min varied from 11.5 to 13 m/s (night) and from 8 to 9.5 m/s (day).

Table 1 presents the averaged (for six-hour period) lidar estimates of the kinetic energy $E$ and the integral scale of turbulence $L_V$ obtained from measurements in daytime for different values of the scan number $N$. It should be noted that the average estimate of the dissipation rate $\varepsilon$ obtained from the same raw lidar data is independent of tabulated $N$ and equal to $4.1 \cdot 10^{-3}$ m$^2$/s$^3$. It follows from Table 1 that as the scan number increases, the estimates of the kinetic energy and the integral scale increase, and for $N > 30$ (measurement time longer than 30 min) the practically complete saturation takes place.

As to the estimates of the turbulence parameters from the nighttime measurement data in the considered period at a height of 200 m, then the averaged (for six-hour period) estimate of the kinetic energy increase linearly with an increase of $N$ from $E = 0.12$ (m/s)$^2$ at $N = 10$ to $E = 0.24$ (m/s)$^2$ at $N = 60$ (twofold increase). The similar increase is also observed for the estimate of the dissipation rate. At $N = 30$, the average estimate $\varepsilon = 5.5 \cdot 10^{-6}$ m$^2$/s$^3$. The integral scale of turbulence determined by Eq. (23) has unrealistically high values ($\sim$ 4 km), which indicates that the above method of lidar data processing is inapplicable to nighttime measurements above the atmospheric surface layer at stable temperature stratification. A possible reason is ignorance of nonstationarity of the average wind, including mesoscale processes (for example, internal gravity waves), at the very weak turbulence. Therefore, we restricted our consideration to the results of lidar measurements of turbulence only in the zone of intense mixing, which occurred in daytime. During the experiment, the intense mixing in the entire layer up to 500 m was observed approximately from 10:30 to 19:00 (7/21/2016), from 11:00 to 20:00 (7/22/2016), and from 11:30 to 18:00 (7/23/2016) LT (Smalikho and Banakh, 2017).

Figure 5 exemplifies the data of lidar measurements at different height. The value of $M^{-1} \sum\limits_{m=0}^{M-1} [< V_{\mathrm{L}}(\theta_m, R_k) > - \mathbf{S}(\theta_m) \cdot < \mathbf{V}(h_k) >]^2$ is more than an order of magnitude smaller than the variance of radial velocity averaged over azimuth angles $\bar{\sigma}_{\mathrm{L}}^2$. The blue curves in Figs.5 (b, d) were obtained with the use of smoothing averaging over three points (azimuth angles). It can be seen that at negative values of the average radial velocity $< V_{\mathrm{L}}(\theta_m, R_k) >$ (or $\mathbf{S}(\theta_m) \cdot < \mathbf{V}(h_k) >$) the variances of the lidar estimate of the radial velocity $\sigma_{\mathrm{L}}^2(\theta_m, h_k)$ mostly exceed the corresponding variances at the positive values of the average radial velocity. As a result, the estimates of the along-wind momentum flux $< uw >$ determined by Eq. (24) (real part) are negative, as expected (Lumley and Panofsky, 1964; Monin and Yaglom, 1971; Byzova et al., 1989; Eberhard et al., 1989; Sathe et al., 2015).

All our results of spatiotemporal visualization of the average wind, turbulence parameters, and instrumental error in estimation of the radial velocity from lidar measurements on July 22 of 2016 in the period under consideration are shown in Fig. 6. Analogous results of estimation of the turbulence parameters were also obtained from lidar measurements on July 21 and 23 of 2016 in the above periods, but on July 23 the wind velocity $U$ was, on average, 1.8 times smaller than that on July





22, while the kinetic energy $E$ was 2 to 2.5 times smaller, and the dissipation rate $\varepsilon$ was 2.5 to 4 times smaller (Smalikho and Banakh, 2017). At the same time, the values of the integral scale $L_V$ were, on average, close to each other.

For illustration, Figs. 7 and 8 show, respectively, the time and height profiles of the wind turbulence parameters and the instrumental error in estimation of the radial velocity. The results presented for $\varepsilon$, $E$, $L_V$, $<uw>$, and $<vw>$ do not contradict the theory of the atmospheric boundary layer and quite correspond to the known experimental data for similar atmospheric conditions (Lumley and Panofsky, 1964; Monin and Yaglom, 1971; Byzova et al., 1989). The instrumental error in estimation of the radial velocity $\sigma_e$ depends mostly on the signal-to-noise ratio SNR : the higher SNR , the smaller $\sigma_e$. Since the probing radiation was focused to a distance of 500 m, $\sigma_e$ took the smallest values in the layer of 200 - 300 m. The error $\sigma_e$ plays an important role in fulfillment of condition (21), when turbulence is very weak. The analysis of results for the kinetic energy of turbulence $E$ has shown that if we ignore the spatial averaging of the radial velocity over the sensing volume (neglect the term $G$ in Eq. (15)), then the obtained estimate of the kinetic energy  is understated by 10 - 20%, especially, in the layer up to 200 m. A necessary condition for obtaining the information about the turbulence energy dissipation rate $\varepsilon$ from lidar data with the use of Eq. (20) is fulfillment of the inequality $R'\psi_l << L_V$. In our case, for heights of 100, 300, and 500 m  at $l = 3$, the separation between the centers of the sensing volumes $R'\psi_3$ is equal, respectively, to 22, 67, and 111 m. According to the data of Fig. 7(d) and Fig. 8(d), this condition is true, that is, the dissipation rate is actually determined within the inertial subrange of turbulence.

The estimation of the integral scale of turbulence $L_V$ by Eq. (23) with the coefficient $C_4 = 0.38$ assumes that the spatial structure of wind turbulence is described by the von Karman model. To clarify how close to reality is this assumption, we have compared the measured azimuth function $\bar{D}_L(\psi_l) - 2\sigma_e^2$ with the function $D_a(\psi_l) = \varepsilon^{2/3} A(l\Delta y_k; L_V)$ , where $A(l\Delta y_k; L_V)$ is calculated by Eq. (19), which takes into account the integral scale of turbulence $L_V$ through replacement of $\Phi(\kappa_1, \kappa_2)$ with

$$\Phi(\kappa_1, \kappa_2; L_V) = \frac{1}{6\pi} \frac{C_1^2 C_2 L_V^{5/3}}{[1 + (C_1 L_V)^2(\kappa_z^2 + \kappa_y^2)]^{4/3}} \left[ 1 + \frac{8}{3} \cdot \frac{(C_1 L_V \kappa_y)^2}{1 + (C_1 L_V)^2(\kappa_z^2 + \kappa_y^2)} \right]. \quad (26)$$

Equation (26) was derived in (Smalikho and Banakh, 2013) with the use of the von Karman model of isotropic turbulence. In calculations of $D_a(\psi_l) = \varepsilon^{2/3} A(l\Delta y_k; L_V)$ , the experimentally obtained values of $\varepsilon$ (from $\bar{D}_L(\psi_l)$ within the inertial subrange of turbulence) and $L_V$ (with the use of Eq. (23)) are used. We have also calculated the degree of deviation of the structure functions $\gamma$ by Eq. (11), where $D_r(l\Delta\theta)$ and $D_\perp(R'l\Delta\theta)$ were substituted with $\bar{D}_L(\psi_l) - 2\sigma_e^2$ and $D_a(\psi_l)$ , respectively.





Figure 9(a) depicts the spatiotemporal distribution of the parameter $\gamma$. According to this figure, the degree of deviation of the structure functions $\gamma$ varies from 0.014 to 0.22 (on average, about 0.1). The widest deviations are observed in the period from 12:30 to 14:30, when the lidar measurements were carried out under convective conditions of the atmospheric boundary layer. Figure 10 exemplifies the comparison of the structure functions $\bar{D}_{\mathrm{L}}(\psi_l)$, $\bar{D}_{\mathrm{L}}(\psi_l) - 2\sigma_e^2$, and $D_a(\psi_l)$. The last example demonstrates the importance of consideration of the instrumental error of the radial velocity in estimation of wind turbulence parameters $\varepsilon$, $E$ и $L_V$. Figures 9(a) and 10 suggest that Eq. (23) with $C_4 = 0.38$ (von Karman model) is applicable to estimation of the integral scale $L_V$. Since turbulence is anisotropic, the estimated integral scale $L_V$ should be considered as the integral scale of turbulence averaged over the azimuth angles of conical scanning at an elevation angle of 35.3°.

To calculate the error of lidar estimates of the dissipation rate, kinetic energy, and the integral scale of turbulence, we used the algorithm of numerical simulation, whose description can be found in papers of Smalikho and Banakh (2013) and Smalikho et al. (2013). In the numerical simulation, we set the input parameters $U$, $\sigma_e$, $\varepsilon$, $E$, and $L_V$ obtained from the lidar experiment. In addition, we assumed the stationarity and statistical homogeneity of the wind field and isotropy of turbulence. Figure 9(b) shows the spatiotemporal distribution of the relative error of lidar estimate of the turbulence energy dissipation rate. The error varies from 6.5% to 15%. Figure 11 shows the time and height profiles of the relative error of estimation of the dissipation rate. It can be seen that for the conditions of this experiment we have the rather high accuracy of determination of the dissipation rate from data of the conically scanning Stream Line lidar. Thus, in the layer of $100 - 350$ m, the relative error does not exceed 7.5%. Worsening of the accuracy of estimation of the dissipation rate with height is caused by an increase of the instrumental error $\sigma_e$ and a decrease of the dissipation rate $\varepsilon$. It is shown in Section 4 that from lidar data measured for four scans it is possible to obtain the estimate of the dissipation rate with a relative error of 20%. The results presented in this section were obtained from the data of 30 scans. An increase in the scan number from 4 to 30 should lead to a decrease of the error from 20% to approximately 7% ($\sqrt{30/4}$ times), which corresponds to the data of Figs. 9(b) and (11) up to a height ~ 350 m.

According to the results of numerical simulation for the experimental conditions considered in this section, the relative error of lidar estimate of the kinetic energy of turbulence varies insignificantly in the time and height ranges of Fig. 6(e) and averages about 10%. At the same time, the relative error of estimation of the integral scale of turbulence varies from 16% to 20% as a function of height and time. A reliable way to study capabilities of the considered method for estimation of the turbulence parameters is comparison of the results of simultaneous measurements by the lidar and the sonic anemometer at the same height.

Section 4 presents the results of simultaneous measurements of the dissipation rate $\varepsilon$ at a height of 43 m by the Stream Line lidar with conical scanning by the probing beam at the elevation angle $\varphi = 9°$ and the sonic anemometer installed at the tower (see Fig. 2). During the lidar measurements, whose results are presented above in Section 5, measurements by the sonic anemometer installed on the tower at a height of 43 m were carried out. Unfortunately, during these measurements the





wind direction was so that the anemometer data were distorted due to wind flow around the tower. On August 27 of 2016, we again conducted joint measurements by the Stream Line lidar (the elevation angle $\varphi$ was also taken equal to 35.3°) and by the sonic anemometer, which measured raw data along the wind without distortions for 24 hours. Since the minimum distance of measurement by the Stream Line lidar is $120-150$ m, it was impossible to conduct lidar measurements at the

anemometer height of 43 m at this elevation angle. Taking into account that the kinetic energy $E$ varies more smoothly with height in comparison with other turbulent parameters $\varepsilon$ and $L_v$, we have compared the diurnal profiles of the kinetic energy obtained from joint measurements by the Stream Line lidar at a height of 100 m and by the sonic anemometer at a height of 43 m. The result of the comparison is shown in Fig. 12. Taking into account the difference in the measurement heights, we can say that a rather good agreement is observed between the time profiles of the kinetic energy of turbulence obtained from

measurements by the different devices.

## 6 Conclusions

Thus, in this paper we have proposed a relatively simple method for determination of the turbulence energy dissipation rate, kinetic energy, and integral scale of turbulence from measurements by conically scanning PCDL. The method is applicable in the case that the longitudinal and transverse dimensions of the sensing volume do not exceed the integral scale of

turbulence. Since the dissipation rate is estimated from the azimuth structure function within the inertial subrange of turbulence, it is sufficient to calculate the function $A(l\Delta y_k)$ for different heights $h_k$ by Eq. (19), and then with Eq. (20) it is possible to retrieve the vertical profiles $\varepsilon(h_k)$. In the estimation of the kinetic energy of turbulence, the spatial averaging of the radial velocity over the sensing volume is taken into account. For this purpose, it is necessary to calculate the function $F(\Delta y_k)$ by Eq. (18) and to use Eq. (15). Then, the integral scale of turbulence is determined with Eq. (23). In contrast to the

approach described by Frehlich et al. (2006) and Smalikho and Banakh (2013), in this method it is not needed to calculate the azimuth structure function of the radial velocity averaged over the sensing volume with the use of the spectrum model in form (26) and to apply the procedure of least-square fitting of the calculated function to the measured one. As was shown in Section 2, this fitting in some cases can lead to the overestimation of the integral scale of turbulence. We have seen this, when applied this fitting to the lidar data of the experiment described in Section 5. As a result, we have obtained

unrealistically high values for estimates of the integral scale of turbulence at low heights of 100–200 m.

     The comparison of measurements of the turbulence energy dissipation rate by the Stream Line lidar with the method described in Section 3 and the data measured by the sonic anemometer has demonstrated a good agreement. The raw data of the lidar experiment of 2016 have been used to obtain the spatiotemporal distributions of different wind turbulence parameters with a height resolution of 10 m and a time resolution of 30 min. The lidar estimates of turbulence have been

analyzed. It has been shown that the use of conical scanning during measurements by PCDL and the method for processing of lidar data proposed in this paper allows obtaining the information about wind turbulence in the atmospheric mixing layer with a rather high accuracy.



*Acknowledgements*. The authors are grateful to colleagues from the Institute of Atmospheric Optics SB RAS A.V. Falits, Yu.A. Rudi, and E.V. Gordeev for the help in experiments.

This study was supported by the Russian Science Foundation, Project No. 14-17-00386.

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



**Table 1**: Average estimates of the kinetic energy and the integral scale of turbulence as functions of the number of scans used during the lidar measurements in the period from 12:00 to 18:00 LT on 22.07.2016 at a height of 200 m.

| Scan number (or measurement duration in min) | 10 | 20 | 30 | 40 | 50 | 60 |
|---|---|---|---|---|---|---|
| Kinetic energy of turbulence, $(m/s)^2$ | 1.71 | 1.84 | 1.88 | 1.91 | 1.92 | 1.93 |
| Integral scale of turbulence, m | 208 | 231 | 239 | 244 | 247 | 249 |





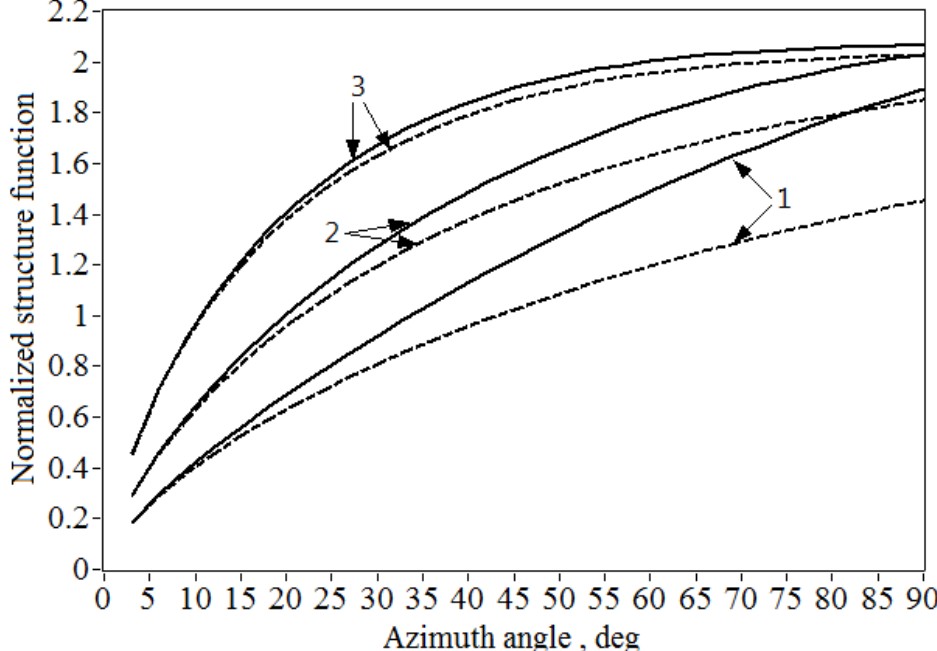

5   **Figure 1**: Normalized structure functions $D_r(\psi)/\sigma_r^2$ (solid curves) and $D_\perp(R'\psi)/\sigma_r^2$ (dashed curves) calculated, respectively, by Eq. (7) and (8) with the use of model (9) at $R'/L_V$ = 0.5 (curves 1), 1 (curves 2), and 2 (curves 3).





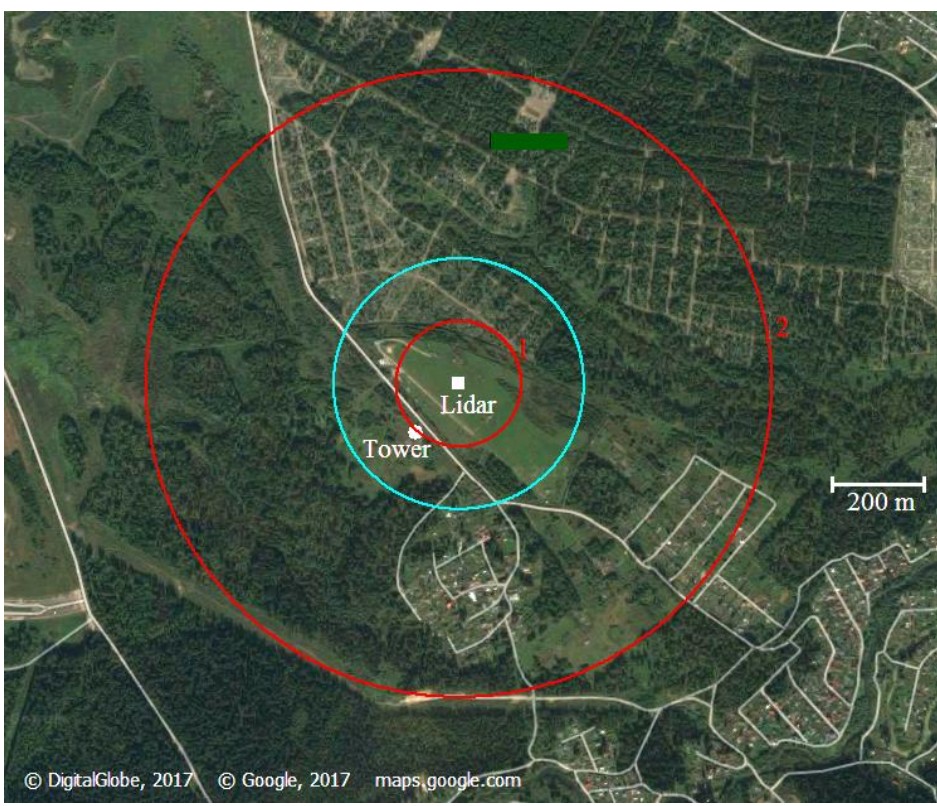

**Figure 2**: Map of the experimental site in 2014 and 2016. The blue circle shows the trajectory of the lidar sensing volume at a height of 43 m during the measurement at the elevation angle $\varphi = 9°$ in 2014. Red circles 1 and 2 shows the trajectories of the lidar sensing volume at heights of, respectively, 100 and 500 m during the measurement at $\varphi = 35.3°$ in 2016. Coordinates of the lidar point were 56°28'51.41"N, 85°06'03.22"E.




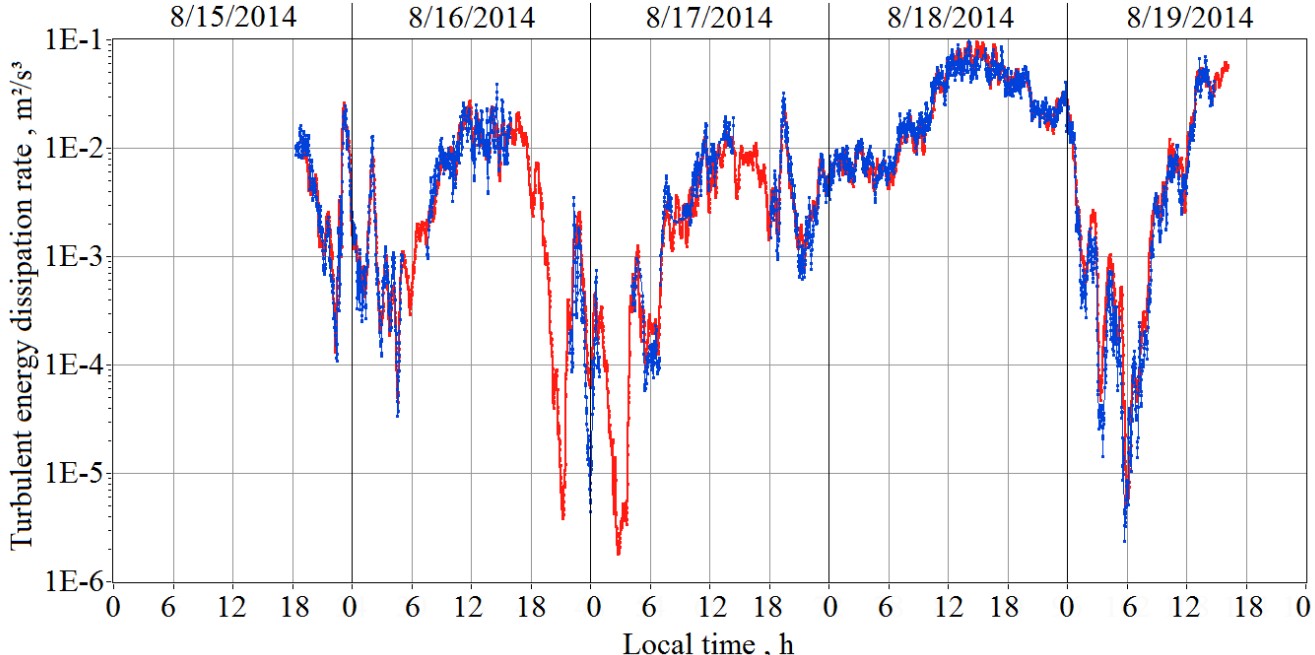

5    **Figure 3**: Time profiles of the turbulence energy dissipation rate obtained from measurements by the sonic anemometer (red curve) and the Stream Line lidar (blue curves) at a height of 43 m.




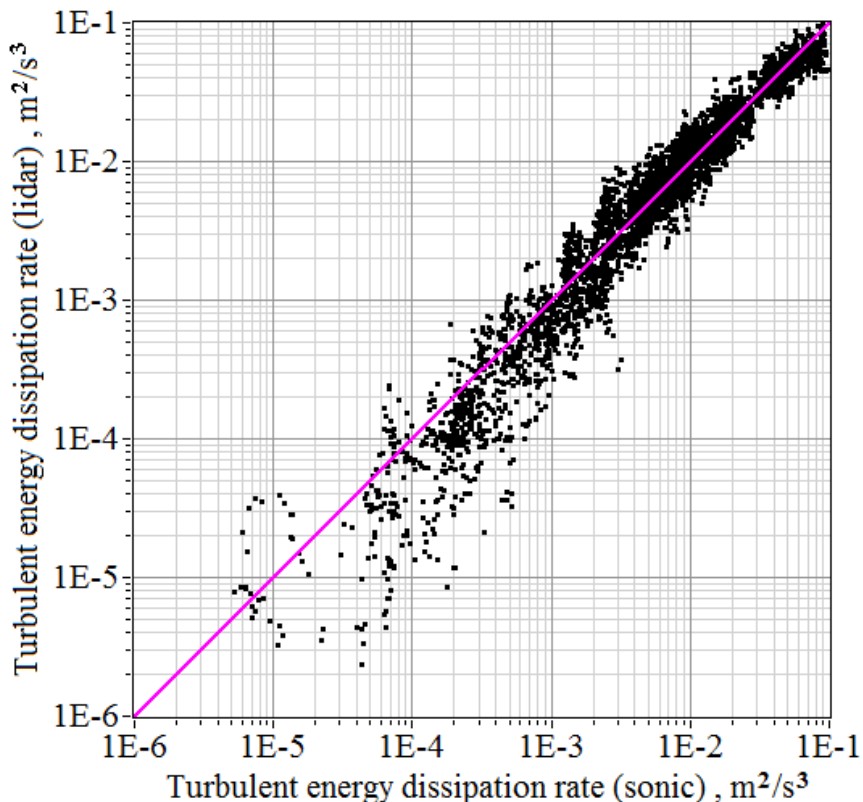

5   **Figure 4**: Comparison of estimates of the turbulence energy dissipation rate obtained from data of simultaneous measurements by the sonic anemometer and the Stream Line lidar. Time profiles of these estimates are shown in Fig.3.





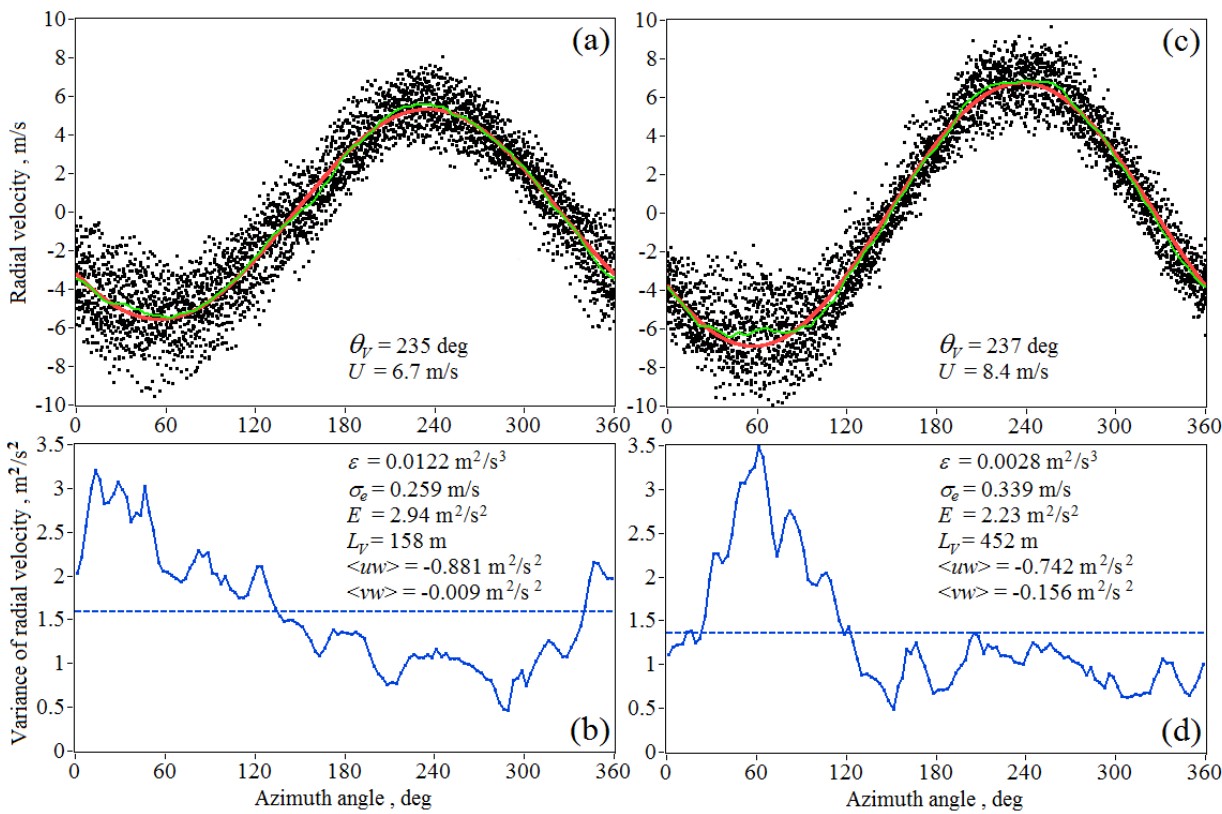

5 **Figure 5**: Single estimates of the radial velocity $V_L(\theta_m, R_k, n)$ (dots); radial velocity averaged over 30 scans, $<V_L(\theta_m, R_k)>$ (green curves); radial velocities as a result of sine-wave fitting, $V_L(\theta_m, R_k) = \mathbf{S}(\theta_m) \cdot <\mathbf{V}(h_k)>$ (red curves) [(a), (c)] and variances of lidar estimate of the radial velocity $\sigma_L^2(\theta_m, h_k)$ (blue curves) [(b), (d)] as functions of the azimuth angle $\theta_m$ obtained from measurements by the Stream Line lidar on 7/22/2016 from 14:09 to 14:39 LT at the heights $h_k = R_k \sin\varphi_E = 109$ m [(a), (b)] and 504 m [(c), (d)]. Dashed curves show the variance averaged over the azimuth angle and the lidar estimate of the radial velocity $\bar{\sigma}_L^2$.





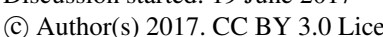

**Figure 6**: Spatiotemporal distributions of the average wind velocity $U$ (a), wind direction angle $\theta_v$ (b), turbulent energy dissipation rate $\varepsilon$ (c), instrumental error of estimation of the radial velocity $\sigma_e$ (d), kinetic energy of turbulence $E$ (e), integral scale of turbulence $L_v$ (f), and momentum fluxes $<uw>$ (g) and $<vw>$ (h) obtained from measurements by the Stream Line lidar on 7/22/2016.





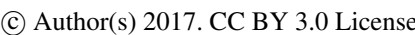

**Figure 7**: Temporal profiles of the turbulent energy dissipation rate $\varepsilon$ (a), instrumental error of estimation of the radial velocity $\sigma_e$ (b), kinetic energy of turbulence $E$ (c), integral scale of turbulence $L_v$ (d), momentum fluxes $<uw>$ (e) and $<vw>$ (f) at heights of 100 m (black curves), 300 m (red curves), and 500 m (blue curves) taken from data of Fig. 6.

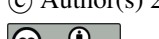



**Figure 8**: Vertical profiles of the turbulent energy dissipation rate $\varepsilon$ (a), instrumental error of estimation of the radial velocity $\sigma_e$ (b), kinetic energy of turbulence $E$ (c), integral scale of turbulence $L_v$ (d), momentum fluxes $<uw>$ (e) and $<vw>$ (f) at 11:30 (black curves), 14:00 (red curves), 17:00 (green curves), and 19:30 (blue curves) taken from the data of Fig. 6.



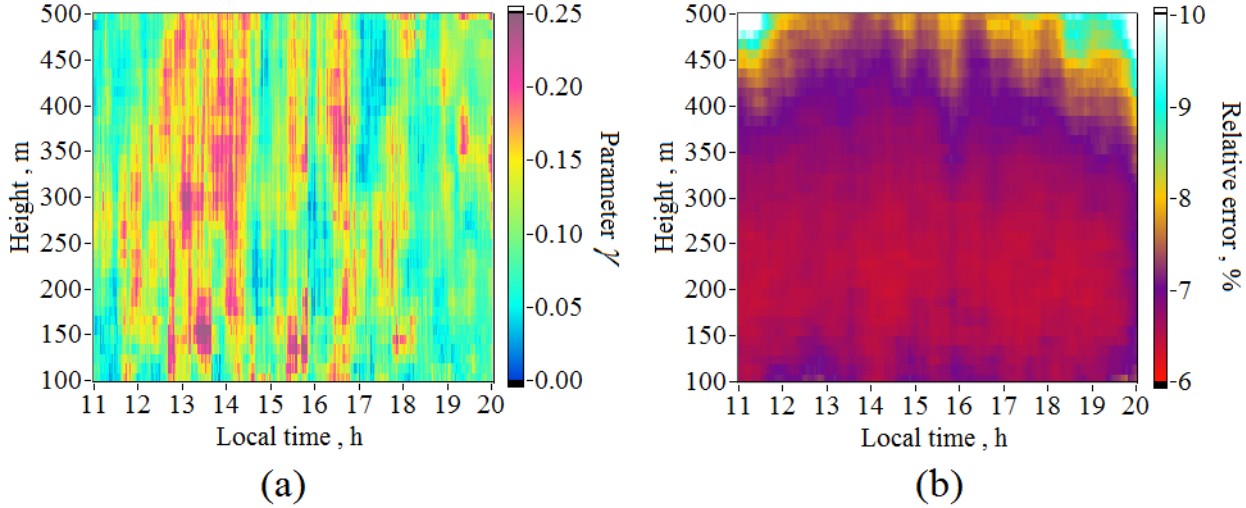

(a)                                                     (b)

5    **Figure 9**: Spatiotemporal distributions of the parameter $\gamma$ (a) and the relative error of estimation of the dissipation rate (b) obtained from measurements by the Stream Line lidar on 7/22/2016.



**Figure 10**: Examples of the azimuth structure functions $\bar{D}_L(\psi_l)$ (green curves), $\bar{D}_L(\psi_l) - 2\sigma_e^2$ (blue curves), and $D_a(\psi_l)$ (red curves) obtained from measurements by the Stream Line lidar on 7/22/2016. The functions $D_a(\psi_l)$ were calculated by Eqs. (17), (19), and (23) with the use of experimental values of $\varepsilon$ and $L_V$. The arc length in the base of the scanning cone $y_k = (\pi/180°)\psi_l h_k / \tan\varphi_E$ at $\psi_l = 90°$ is given in parenthesis.





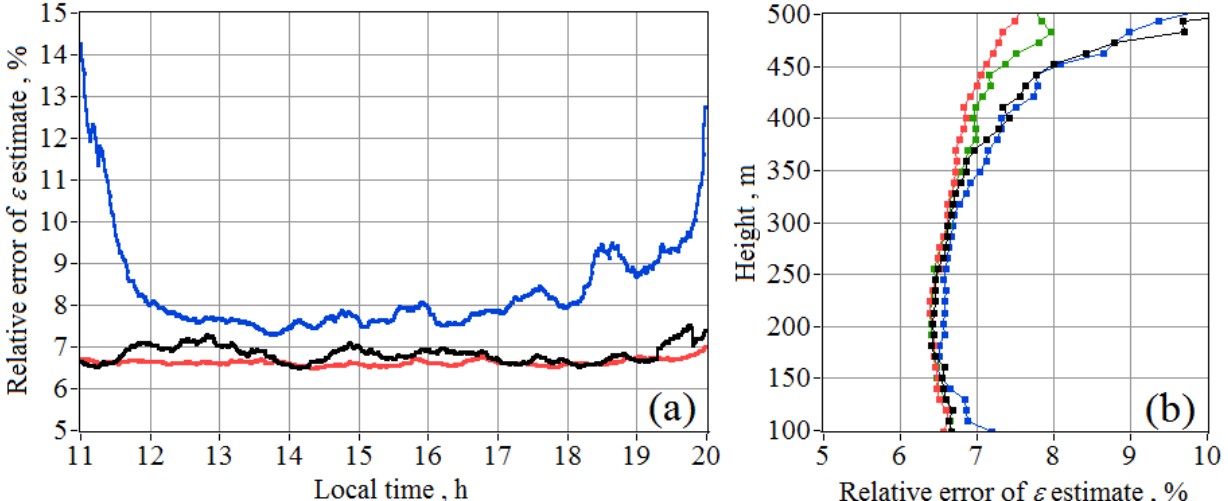

5    **Figure 11**: Time profiles of the relative error of estimation of the turbulence energy dissipation rate  (a) at heights of 100 m (black curve),
300 m (red curve), and 500 m (blue curve) and height profiles of the relative error of estimation of the turbulence energy dissipation rate
(b) at 11:30 (black squares), 14:00 (red squares), 17:00 (green squares), and 19:30 (blue squares) calculated from data of Fig. 8 (a,b).





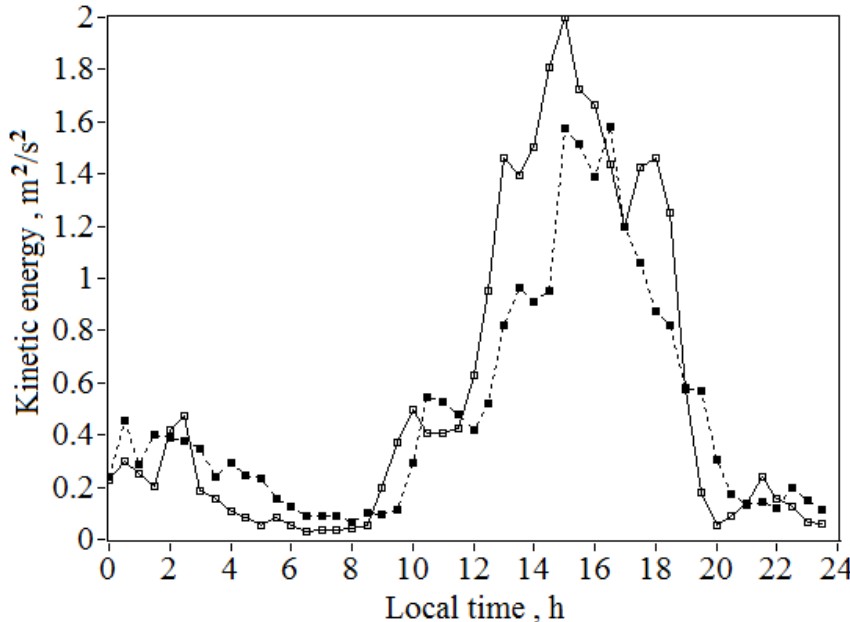

**Figure 12**: Diurnal profiles of the kinetic energy of turbulence obtained from simultaneous measurements by the sonic anemometer at a height of 43 m (squares connected by solid lines) and the Stream Line lidar at a height of 100 m (squares connected by dashed lines) on 8/27/2016.