# Peer review of "Measurements of wind turbulence parameters by a conically scanning coherent Doppler lidar in the atmospheric boundary layer"

_Atmospheric Measurement Techniques, 2017_

## Referee Comment (RC1) · Anonymous Referee #1 · 14 Jul 2017

In this manuscript, the authors describe how various turbulent parameters can be measured with a continuously conically scanning Doppler lidar. The techniques for measurement of the parameters are described in detail, and sample results of the measurements are shown. Doppler lidar measurements of the dissipation rate are compared with a sonic anemometer at 43 m, and are shown to generally agree well, except with some low biases under stable conditions when the lidar is unable to resolve the any portion of the inertial subrange. The turbulence kinetic energy from the Doppler lidar is shown to generally agree with measurements from a sonic anemometer at a lower

height. In all, the scientific quality of the manuscript appears to be solidly based in theory and good. The work builds on previous work, with new refinements made to the strategy. However, there are a few areas of the manuscript that could be clarified, as sections of the text are difficult to follow. As such, I recommend this manuscript be suitable for publication in AMT after minor revisions, in which the following comments, which are mostly of clarification, are addressed.

Specific Comments:

a) P. 1, line 19; p. 2, line 5 (and elsewhere): Change 'raw lidar data' to 'radial velocities'. By 'raw data', I interpret that to be the measured Doppler spectrum, which are not used directly in the referenced techniques to measure turbulence.

b) p. 2, line 9: By 'averaging over the sensing volume', clarify that you mean the spatial-temporal averaging of the pulse length over one beam accumulation and not the averaging over the entire conical area.

c) p. 2 line 12: What are dr and $\sigma r$?

d) p. 2 line 20: Quantify 'high spatial resolution'.

e) p. 2 line 23: What disadvantages of the earlier methods, precisely? The averaging over the sensing volume?

f) p. 2 line 24: Change 'spatiotemporal' to 'time and height'. The term 'spatiotemporal' is too general, and generally means that information on the horizontal variability is measured/known.

g) p. 6 lines 22-24: This section is difficult to follow. Providing more text to describe the different terms and how they are related would be helpful.

h) p. 7-8: For this section in particular, it would be helpful to add a figure providing a few examples of the 2-dimensional spectrum and showing how the different parameters are calculated from it (particularly interested in $\sigma e$, $\sigma t$), including adding a paragraph

discussing the figures. This would be similar to showing how different parameters are calculated in Fig. 5.

i) p. 10 line 10: How much of the data was unusable exactly? The percentage of unusable data would be helpful.

j) p. 10 line 13: What was the averaging time that the results shown in Fig. 3 were computed over? Based on p. 9 lines 19/24, it seems that 4 PPIs were used (over 5 minutes) while the sonic anemometer used 20 min of data. How were these differences in averaging times rectified?

k) p. 12 line 5: Is it possible to discern that the increase in kinetic energy computed over more scans (over longer time periods) is truly a better measure, and not simply due to non-stationarity of the mean wind (as discussed for the stable case at line 15) increasing the variances across the entire conical scan? Based on Fig. 6, the mean wind changes (wind speed slowly decreases, direction shifts) over the 6 hour time window mentioned, thus this may be causing the increase in measured TKE.

l) p. 12 line 15: Other possible reasons include the inability of the lidar to resolve any portion of the inertial subrange (thus all derived parameters are not valid) and the low bias of dissipation (denominator for calculation of integral scale) when it is small.

m) p. 12 line 20-22: The meaning and significance of 'The value of . . . over azimuth angles' is unclear; it should be rewritten.

n) p. 13 line 2: What is meant by 'close to each other'? A quantitative measure (standard deviation or range of values) is needed.

o) p. 15 line 125: Add the qualifier here that these high estimates were under stably stratified conditions.

p) End of manuscript: With the large number of variables and subscripts in this manuscript, adding a list of the symbols would be extremely helpful in reading this manuscript. I had to keep searching through the paper to find variables that were first

introduced many pages earlier in the paper.

Technical corrections:

a) p. 6 line 10 (and reference list): 'Pearson' not 'Pierson'

b) p. 6 line 20: Should $\sigma e2$ have an overbar as well?

c) P. 11 line 15: 'continuously' is a better word than 'permanently'

---

## Referee Comment (RC2) · Anonymous Referee #2 · 22 Aug 2017

Review for Manuscript ID: AMT-2017-140

Title: Measurements of wind turbulence parameters by a conically scanning coherent Doppler lidar in the atmospheric boundary layer

Authors: Igor N. Smalikho, Viktor A. Banakh

General comments ——————-

This manuscript presents a methodology for deriving turbulent parameters from scanning Doppler lidar observations in the lower atmosphere. The methodology is sound

and the results show that the parameters derived from Doppler lidar measurements usually agree well with reference parameters obtained from a sonic anemometer.

The methodology uses a particular turbulence model which dictates how certain properties of the observed turbulence are expected to behave and so enable them to be derived. A clear statement describing atmospheric situations when this model is applicable, and situations when it is not likely to be applicable, should be included in the conclusion. Are there methods for checking whether the turbulence model is applicable in a particular situation? For example, can you use the Doppler lidar observations to check for stationarity? In addition, what are the likely biases if the model is not strictly applicable, but provides reasonable results? An example here is the slight underestimates in turbulent energy dissipation rate provided by the Doppler lidar at low values. Is this expected because of unrealistic integral scales used, or is it an issue in accounting for radial velocity measurement uncertainty correctly?

The manuscript contains all of the information necessary for publication, but in its current state is difficult to read. There are a huge number of variables and subscripts introduced, which although necessary for completeness, make it difficult to follow. It would be easier to comprehend if large parts of the derivation were placed in an appendix, with terms directly related to the parameters that will be derived from observations included in the text. In addition, the instrument should be introduced first in Section 3, so that it is easy to refer to the instrument specifications when introducing the measurement strategy. Add a table presenting the relevant instrument specifications, e.g. pulse-repetition-frequency, receiver bandwidth/Nyquist velocity, range gate length, azimuthal scanning speed, lidar wavelength, telescope type, rather than referring the reader to another paper. As an aid to the reader, this table could also include the associated variable in the equations.

After some minor modifications, I feel this manuscript will be suitable for publication.

Specific comments ——————-

Page 1, line 19: The data provided by these instruments is not really 'raw' data, but radial velocities.

Page 2, line 13: Suggest replacing 'were proposed' by 'have been proposed'.

Page 2, line 24: Need to state that this is '100 to 500 m in altitude', as it could be assumed that the distances refer to range.

Page 2, line 27: Suggest starting the paragraph with 'First, we describe the equations that will be used to develop the measurement strategy and method for deriving the wind turbulence parameters:'

Page 2, line 28: The measured 'raw' radial velocities are not strictly instantaneous, as they are obtained by averaging a large number of samples internally.

Page 4, line 4: Suggest replacing 'some or other' with 'an appropriate'.

Page 4, lines 8-9: It would be clearer for the reader if these expressions were placed on separate lines.

Page 3, line 15; page4, lines 20-24; and Figure 1: It should be made clear, especially in the Figure caption, that the azimuth angle refers to the azimuthal resolution (if continuous scan) or separation between 2 adjacent rays in a scan (step-stare scan).

Page 5, line 1: Suggest replacing 'the both' with 'both'.

Page 5, line 5: What is the rationale behind choosing delta theta = 3 degrees? And what does L correspond to?

Page 5, Measurement strategy: Do you mean that you perform one conical scan with +ve azimuth rotation, then one scan with -ave azimuth rotation?

Page 5, line 24: As defined previously, R_0 should be (delta R / 2) if the first range gate is k=0, unless you define k=0 as the first usable range gate. Then 'minimal distance' should be defined precisely, e.g. define 'R_0 is the distance to the first usable range

gate' before the equation on line 23, and explain why the first gate should satisfy the condition stated on line 25.

Page 5, line 26: The maximum range is effectively determined by the instrument pulse repetition frequency; the maximum usable range depends on the signal-to-noise ratio (SNR) and hence the atmosphere. Suggest rewriting this sentence, stating instead that the 'uncertainty in the radial velocity measurement depends on the SNR'.

Page 6, line 9: Use correct reference (Pearson)

Page 6, line 11: Do you mean azimuthal dimension rather than longitudinal dimension here?

Page 6, line 14: How do you know if Lv only occasionally exceeds the sensing volume?

Page 6, lines 15-17: Other authors have shown that it is usually safer to always take account of the uncertainty in the radial velocity estimates.

Page 6, lines 18-24: This sequence of equations requires much more explanation than is given here.

?? Page 8, lines 12-15: Not sure that this can be justified without evidence..

Page 9, line 16, and page 11, line 15: The focus of the lidar beam was set to XX m.

Page 11, line 9: Suggest 'To test the method for determining the kinetic energy,..'

Page 11, line 12-15: Suggest 'The presence of forest fires in the Tomsk region provided lidar measurements with high signal-to-noise ratios ...'

Page 11, line 15: Suggest replacing 'permanently' with 'continuously'.

Page 11, line 20: The 'minimum useful range'.

Page 12, line 3: I assume you mean 'horizontal wind speed'.

Page 14, line 21: This assumes that the turbulent parameters don't change over the

time required to obtain 30 scans.

Figure 3: Suggest replacing 'Time profiles of the turbulence' with 'Time series of the turbulent'.

Figure 4: Suggest replacing 'Time profiles' with 'Time series'.

Figure 6: Panel (a) should state 'Wind speed' rather than 'Wind velocity' for the colorbar title.

Figure 7: Suggest replacing 'Temporal profiles' with 'Time series',

Figure 7,8 Suggest replacing 'instrumental error of estimation of the radial velocity' with 'uncertainty in radial velocity estimate'.

Figure 9: Suggest replacing 'Spatiotemporal distributions' with 'Time-height plots', and 'relative error of estimation of the dissipation rate' with 'relative error in dissipation rate'.
* * *

---

## Author Comment (AC1) · 11 Sep 2017

Referee #1 In this manuscript, the authors describe how various turbulent parameters can be measured with a continuously conically scanning Doppler lidar. The techniques for measurement of the parameters are described in detail, and sample results of the measurements are shown. Doppler lidar measurements of the dissipation rate are compared with a sonic anemometer at 43 m, and are shown to generally agree well, except with some low biases under stable conditions when the lidar is unable to resolve the any portion of the inertial subrange. The turbulence kinetic energy from the Doppler

lidar is shown to generally agree with measurements from a sonic anemometer at a lower height. In all, the scientific quality of the manuscript appears to be solidly based in theory and good. The work builds on previous work, with new refinements made to the strategy. However, there are a few areas of the manuscript that could be clarified, as sections of the text are difficult to follow. As such, I recommend this manuscript be suitable for publication in AMT after minor revisions, in which the following comments, which are mostly of clarification, are addressed. Specific Comments: a) P. 1, line 19; p. 2, line 5 (and elsewhere): Change 'raw lidar data' to 'radial velocities'. By 'raw data', I interpret that to be the measured Doppler spectrum, which are not used directly in the referenced techniques to measure turbulence. => The phrase "raw data measured" has been replaced by "measurements".

b) p. 2, line 9: By 'averaging over the sensing volume', clarify that you mean the spatial-temporal averaging of the pulse length over one beam accumulation and not the averaging over the entire conical area. => Page 2, line 9 : "(see Eq.(6) in paper of Smalikho and Banakh, 2013)" has been added.

c) p. 2 line 12: What are dr and $\sigma r$? => Page 2, line 13: "and .. is the variance" and "radial" have been added. "dr" is an infinitesimal increment of the integration variable "r" (separation between two points).

d) p. 2 line 20: Quantify 'high spatial resolution'. => Page 2, lines 21-22: "(longitudinal size of the sensing volume can be around 30 m)" has been added.

e) p. 2 line 23: What disadvantages of the earlier methods, precisely? The averaging over the sensing volume? => We do not know publications in which authors would take into account the effect of averaging of the radial velocity over the sensing volume when estimating the kinetic energy of turbulence.

f) p. 2 line 24: Change 'spatiotemporal' to 'time and height'. The term 'spatiotemporal' is too general, and generally means that information on the horizontal variability is measured/known. => Fixed.

g) p. 6 lines 22-24: This section is difficult to follow. Providing more text to describe the different terms and how they are related would be helpful. => Text on page 6 (lines 18-24) of initial version of the manuscript has been replaced by the text on page 6 (lines 19-26) and page 7 (lines 1-3) of the revised manuscript. Page 7, line 13: "(Banakh and Smalikho, 2013)" has been added.

h) p. 7-8: For this section in particular, it would be helpful to add a figure providing a few examples of the 2-dimensional spectrum and showing how the different parameters are calculated from it (particularly interested in $\sigma$e, $\sigma$t), including adding a paragraph discussing the figures. This would be similar to showing how different parameters are calculated in Fig. 5. => Page 9, lines 3-13: The paragraph "With increasing range . . . without taking into account the averaging of the radial velocity over the sensing volume." has been added. Page 9, lines 14-26: The paragraph "Figure 2 shows vertical profiles . . . the underestimation does not exceed 5%." has been added. Page 24: Figure 2 has been added. The sentence "The analysis of results for the kinetic energy of turbulence . . . is understated by 10 - 20%, especially, in the layer up to 200 m." (page 13, lines 9-12 in the initial variant of the manuscript) has been removed. Page 14, lines 22-30 and page 15, lines 1-4: The paragraph "Under the condition . . . then underestimation of the integral scale will be from 15% to 40%." has been added. Page 32: Figure 10 has been added.

i) p. 10 line 10: How much of the data was unusable exactly? The percentage of unusable data would be helpful. => Page 11, line 16: "(around 15%)" has been added.

j) p. 10 line 13: What was the averaging time that the results shown in Fig. 3 were computed over? Based on p. 9 lines 19/24, it seems that 4 PPIs were used (over 5 minutes) while the sonic anemometer used 20 min of data. How were these differences in averaging times rectified? => If the same measurement time is used for the lidar and the sonic anemometer, the distance traveled by the sensing volume and the distance to which the air masses are carried by the mean wind during this time will vary greatly, since the velocity of the mean wind is substantially less than the linear velocity of

movement of the sensing volume at the base of the scanning cone. We believe that in order to compare the results of estimating the dissipation rate, it is more appropriate to use the lidar data and the acoustic anemometer data, which correspond to the same distances.

k) p. 12 line 5: Is it possible to discern that the increase in kinetic energy computed over more scans (over longer time periods) is truly a better measure, and not simply due to non-stationarity of the mean wind (as discussed for the stable case at line 15) increasing the variances across the entire conical scan? Based on Fig. 6, the mean wind changes (wind speed slowly decreases, direction shifts) over the 6 hour time window mentioned, thus this may be causing the increase in measured TKE. => The variance of the average (30-minute averaging) of the wind velocity, calculated from the data in Figure 6 (a) for a height of 200 m and a time interval from 12:00 to 18:00, is approximately 10 times less than the TKE given in Table 1 (for 30 scans). Therefore, we can assume that the contribution of the nonstationarity of the mean wind to the kinetic energy estimate is negligible, in comparison with the turbulent fluctuations of the wind field. However, for another case considered in the manuscript (measurement at an altitude of 200 m from 01:00 to 07:00), the variance of the average (30-minute averaging) of the wind velocity is approximately twice the estimate of the kinetic energy obtained by using lidar data for 30 scans. This is the reason that, with an increase in the averaging interval from 10 min to 60 min, the magnitude of the kinetic energy estimate is monotonically increasing (it has no saturation, as in the first case under consideration). Apparently, for conditions of very weak turbulence on the background of nonstationarity of the mean wind, a special procedure for data filtering is required, which is not the subject of this paper.

l) p. 12 line 15: Other possible reasons include the inability of the lidar to resolve any portion of the inertial subrange (thus all derived parameters are not valid) and the low bias of dissipation (denominator for calculation of integral scale) when it is small. => We agree with this comment. Under conditions of stable thermal stratification of the

atmosphere, the inertial subrange of turbulence can be much smaller than the size of the sensing volume, or even the inertial interval may be absent. It is obvious that the method of estimating the dissipation rate and the integral scale described in the manuscript is not applicable for this case. Therefore, in this manuscript there are no results of data processing, measured by the lidar in 2016 at night.

m) p. 12 line 20-22: The meaning and significance of 'The value of . . . over azimuth angles' is unclear; it should be rewritten. => Fixed.

n) p. 13 line 2: What is meant by 'close to each other'? A quantitative measure (standard deviation or range of values) is needed. => Page 14, lines 9-10: "(maximum deviation is around 20%)" has been added.

o) p. 15 line 125: Add the qualifier here that these high estimates were under stably stratified conditions. => Probably, the reviewer has in mind line 25. Page 17, line 14: "(measurements in the daytime)" has been added. Page 17, line 16: the sentence "Sometimes such estimates exceed 1 km in contrast to results shown in Figures 6(f), 7(d) and 8(d)." has been added.

p) End of manuscript: With the large number of variables and subscripts in this manuscript, adding a list of the symbols would be extremely helpful in reading this manuscript. I had to keep searching through the paper to find variables that were first introduced many pages earlier in the paper. => Pages 17-20: Appendix with a list of symbols has been added.

Technical corrections: a) p. 6 line 10 (and reference list): 'Pearson' not 'Pierson' => Fixed.

b) p. 6 line 20: Should $\sigma e2$ have an overbar as well? => Page 6, lines 25-26: The sentence "In Eqs. (13) and (14) it is assumed that. . . is independent of the azimuth angle . . ." has been added.

c) P. 11 line 15: 'continuously' is a better word than 'permanently'. => Fixed.

Please also note the supplement to this comment:
https://www.atmos-meas-tech-discuss.net/amt-2017-140/amt-2017-140-AC1-
supplement.pdf

**Supplement:**

[revised manuscript text omitted]

±19,4 m/s for the Stream Line lidar), regardless of the true value of the velocity, can significantly differ from zero. To avoid the application of the data filtering procedure, the measured array $V_L(\theta_m, R_k, n)$ must not contain "bad" estimates. Then, the lidar estimate of the radial velocity can be represented as (Frehlich and Cornman, 2002; Banakh and Smalikho, 2013)

$$V_L(\theta_m, R_k, n) = V_a(\theta_m, R_k, n) + V_e(\theta_m, R_k, n),$$ (12)

5     where $V_a(\theta_m)$ is the radial velocity averaged over the sensing volume with the longitudinal dimension $\Delta z$ and the transverse dimension $\Delta y_k = \Delta\theta R_k \cos\varphi_E$ (here, $\Delta\theta$ is in radians), and $V_e(\theta_m)$ is the random instrumental error of estimation of the radial velocity having the following properties: $<V_e> = <V_a V_e> = 0$ and $<V_e(\theta_m)V_e(\theta_l)> = \sigma_e^2 \delta_{m-l}$ ($\sigma_e^2$ is the variance of random error, $\delta_m$ is the Kronecker delta). For the conditions of stationary and homogeneous turbulence, the estimate is unbiased, that is, $<V_L(\theta_m, R_k, n)> = <V_r(\theta_m, R_k)>$.

10     Lidars of Stream Line type, one of which is used in our experiments, are characterized by formation of a sensing volume of relatively small size, for example, with the longitudinal dimension $\Delta z = 30$ m (Pearson et al., 2009). When the conical scanning with $\varphi = \varphi_E$ and $\Delta\theta = 3°$ is used, the transverse dimension of the sensing volume increases linearly from 8.5 m at $R_k = 200$ m to 42.8 m at $R_k = 1$ km. It is important to take into account the effect from averaging of the radial velocity over the sensing volume not only when estimating the dissipation rate $\varepsilon$ within the inertial subrange of turbulence, but also when

15     estimating the parameters $E$ and $L_V$, especially, when $L_V$ exceeds the size of the sensing volume insignificantly. Even at the high signal-to-noise ratio and the large number of probing pulses used for accumulation of lidar data, when the variance $\sigma_e^2$ is extremely small, it is necessary to take into account the instrumental error of estimation of the radial velocity, if turbulence is very weak (Frehlich et al., 2006).

    After the corresponding manipulations, from Eq. (12), taking into account statistical properties of the random error

20     $V_e(\theta_m)$, we derived the following equations for the variance and the structure function of lidar estimate of the radial velocity averaged over all azimuth angles $\theta_m$:

$$\bar{\sigma}_L^2 = \bar{\sigma}_a^2 + \sigma_e^2,$$ (13)

$$\bar{D}_L(\psi_l) = \bar{D}_a(\psi_l) + 2\sigma_e^2,$$ (14)

    where $\quad \bar{\sigma}_\alpha^2 = M^{-1}\sum_{m=0}^{M-1}\sigma_\alpha^2(\theta_m) \quad ; \quad \sigma_\alpha^2(\theta_m) = <[V_\alpha'(\theta_m)]^2> \quad ; \quad \bar{D}_\alpha(\psi_l) = (M-l)^{-1}\sum_{m=0}^{M-1-l}D_\alpha(\psi_l, \theta_m) \quad ; \quad D_\alpha(\psi_l, \theta_m) =$

25     $<[V_\alpha'(\theta_m + \psi_l) - V_\alpha'(\theta_m)]^2>; \quad V_\alpha' = V_\alpha - <V_r>$ and subscript $\alpha$ is L or $a$. In Eqs. (13) and (14) it is assumed that $\sigma_e$ is independent of the azimuth angle $\theta_m$. The variance $\bar{\sigma}_a^2$ can be represented as

$$\bar{\sigma}_a^2 = \bar{\sigma}_r^2 - \bar{\sigma}_t^2 , \tag{15}$$

where $\bar{\sigma}_t^2 = M^{-1} \sum_{m=0}^{M-1} \sigma_t^2(\theta_m)$ and $\sigma_t^2(\theta_m) = \sigma_r^2(\theta_m) - \sigma_a^2(\theta_m)$ is turbulent broadening of the Doppler spectrum (Banakh and Smalikho, 2013).

Having specified the high resolution in the azimuth angle (large number $M$) and $\varphi = \varphi_E$, from Eqs. (13) - (15) with allowance made for Eq. (4), we obtain the equation for the kinetic energy of turbulence in the form

$$E = (3/2)[\bar{\sigma}_L^2 - \bar{D}_L(\psi_1)/2 + G], \tag{16}$$

where $G = \bar{\sigma}_t^2 + \bar{D}_a(\psi_1)/2$. At $L_V > \max\{\Delta z, \Delta y_k\}$, the dimensions of the sensing volume do not exceed the low-frequency boundary of the inertial subrange, for which turbulence is locally isotropic and, correspondingly, $G \sim \varepsilon^{2/3}$. If the condition $l\Delta y_k << L_V$ is additionally fulfilled, then for calculation of the turbulent broadening of the Doppler spectrum $\bar{\sigma}_t^2 = \sigma_t^2$ and the structure function $\bar{D}_a(\psi_l) = D_a(\psi_l)$ we can use the two-dimensional spatial Kolmogorov—Obukhov spectrum. For these conditions, the Gaussian temporal profile of the probing pulse, and the rectangular time window used for obtaining of Doppler spectra, we have derived the following equations (Banakh and Smalikho, 2013):

$$\sigma_t^2 = \varepsilon^{2/3} F(\Delta y_k) , \tag{17}$$

$$D_a(\psi_l) = \varepsilon^{2/3} A(l\Delta y_k) . \tag{18}$$

In Eqs. (17) and (18)

$$F(\Delta y_k) = \int_0^\infty d\kappa_1 \int_0^\infty d\kappa_2 \Phi(\kappa_1, \kappa_2)[1 - H_\parallel(\kappa_1) H_\perp(\kappa_2)] , \tag{19}$$

$$A(l\Delta y_k) = 2\int_0^\infty d\kappa_1 \int_0^\infty d\kappa_2 \Phi(\kappa_1, \kappa_2) H_\parallel(\kappa_1) H_\perp(\kappa_2)[1 - \cos(2\pi l\Delta y_k \kappa_2)] , \tag{20}$$

where $\Phi(\kappa_1, \kappa_2) = C_3(\kappa_1^2 + \kappa_2^2)^{-4/3}[1 + (8/3)\kappa_2^2/(\kappa_1^2 + \kappa_2^2)]$ ; $C_3 = 2C_2/(3\pi C_1^{2/3})$ = 0.0652; $H_\parallel(\kappa_1) = [\exp\{-(\pi\Delta p \kappa_1)^2\} \operatorname{sinc}(\pi\Delta R \kappa_1)]^2$ is the longitudinal transfer function of the low-frequency filter, and $H_\perp(\kappa_2) = [\operatorname{sinc}(\pi\Delta y_k \kappa_2)]^2$ is the transverse one; $\Delta p = c\sigma_p/2$ ; $c$ is the speed of light; $2\sigma_p$ is the duration of the probing pulse determined by the $e^{-1}$ power level to right and to the left from the peak point, $\Delta R = cT_W/2$ , $T_W$ is the temporal window width; and $\operatorname{sinc}(x) = \sin x/x$ .

In Eq. (16), $\bar{\sigma}_L^2$ and $\bar{D}_L(\psi_1)$ are directly determined from experimental data. To take into account the term $G = \varepsilon^{2/3}[F(\Delta y_k) + A(\Delta y_k)/2]$ in Eq. (16), it is necessary to have information about the dissipation rate $\varepsilon$. According to

Eq. (14), the difference $\bar{D}_\mathrm{L}(\psi_l) - \bar{D}_\mathrm{L}(\psi_1)$ is equal to the difference $\bar{D}_a(\psi_l) - \bar{D}_a(\psi_1)$. Within the framework of the above conditions and according to Eq. (18), the latter is equal to $\varepsilon^{2/3}[A(l\Delta y_k) - A(\Delta y_k)]$. Then the dissipation rate can be determined as

$$\varepsilon = \left[ \frac{\bar{D}_\mathrm{L}(\psi_l) - \bar{D}_\mathrm{L}(\psi_1)}{A(l\Delta y_k) - A(\Delta y_k)} \right]^{3/2}, \tag{21}$$

5  where the number $l > 1$ should be so that, on the one hand, the consideration is within the inertial subrange and, on the other hand, the condition

$$[\bar{D}_\mathrm{L}(\psi_l) - \bar{D}_\mathrm{L}(\psi_1)] \gg \bar{D}_\mathrm{L}(\psi_1)\sqrt{2/(MN)} \tag{22}$$

is fulfilled. This condition provides for the high accuracy of estimation of the dissipation rate at the large numbers $M$ and $N$. In parallel, we can calculate the instrumental error of estimation of the radial velocity $\sigma_e$ as

10  $$\sigma_e = \sqrt{[\bar{D}_\mathrm{L}(\psi_1) - \varepsilon^{2/3}A(\Delta y_k)]/2} \equiv \sqrt{\frac{\bar{D}_\mathrm{L}(\psi_1)A(l\Delta y_k) - \bar{D}_\mathrm{L}(\psi_l)A(\Delta y_k)}{2[A(l\Delta y_k) - A(\Delta y_k)]}}. \tag{23}$$

Using the lidar estimates of the kinetic energy $E$ (by Eq. (16)) and the dissipation rate $\varepsilon$ from experimental data, we can determine the integral scale $L_V$ by Eqs. (4) and (10) as

$$L_V = C_4 E^{3/2}/\varepsilon, \tag{24}$$

where $C_4 = [2/(3C_2)]^{3/2} = 0.3796$.

15  Taking into account that the elevation angle $\varphi = \varphi_E = \tan^{-1}\left(1/\sqrt{2}\right)$, we use the following equation (Eberhard et al., 1989) for determination of the momentum fluxes $$ and $<vw>$:

$$ + j<vw> = \frac{3}{\sqrt{2}} \frac{1}{M} \sum_{m=0}^{M-1} \sigma_\mathrm{L}^2(\theta_m) \exp[j(\theta_m - \theta_V)], \tag{25}$$

where $j = \sqrt{-1}$. Since the instrumental error of estimation of the radial velocity $\sigma_e$ is independent of the azimuth angle $\theta_m$ and within the sensing volume, turbulence is locally isotropic (the condition $L_V > \max\{\Delta z, \Delta y_k\}$ is assumed to be true), that

20  is $\sigma_t^2$ does not depend on $\theta_m$, it is not necessary here to take into account the instrumental error and the effect from averaging of the radial velocity over the sensing volume. Indeed, as shown by Eberhard et al. (1989), in the case of a horizontally homogeneous turbulence statistics and very large $M$, equation (25) is exact, if $\sigma_\mathrm{L}^2(\theta_m)$ is replaced by $\sigma_r^2(\theta_m)$.

On the other hand, $\sigma_L^2(\theta_m) = \sigma_r^2(\theta_m) - \sigma_t^2 + \sigma_e^2$. Taking into account that $\sigma_t^2$ and $\sigma_e^2$ do not depend on $\theta_m$ and

$$\frac{1}{M}\sum_{m=0}^{M-1}\exp[\,j(\theta_m - \theta_V)] = 0\,,$$ Eq. (25) can also be regarded as exact.

With increasing range $R_k = R_0 + k\Delta R$, the measurement height $h_k = R_k \sin\varphi$ and the transverse dimension of the sensing volume $\Delta y_k = \Delta\theta R_k \cos\varphi$ increase linearly. Using Eqs. (19) and (20), we calculated $F(\Delta y_k)$ and $A(l\Delta y_k)$ by specifying the parameters of the lidar experiment conducted in 2016 (see Section 5), that is, $\varphi = \varphi_E = 35.3°$, $\Delta\theta = \pi/60$ (3°), $\Delta R = 18$ m and $\Delta p = 15.3$ m. Without taking into account the spatial averaging of the radial velocity over the sensing volume, in Eq. (20) we set $H_{\parallel}(\kappa_1) = H_{\perp}(\kappa_2) = 1$ and $A(l\Delta y_k) = A_0(l\Delta y_k)$. Then, after integrating over $\kappa_1$ and $\kappa_2$ in Eq. (20), we obtain the following equation: $A_0(l\Delta y_k) = 2.667(l\Delta y_k)^{2/3} = (4/3)C_K(l\Delta y_k)^{2/3}$. According to Fig. 1, the azimuth and transverse structure functions of the radial velocity completely coincide under the condition $l\Delta\theta \le 9°$. Therefore, we carried out calculations of $A(l\Delta y_k)$ at $l = 1$ and $l = 3$. To estimate the turbulent energy dissipation rate by equation (21), we set $l = 3$. Denote by $\varepsilon_0$ the dissipation rate estimate obtained after the replacement of the difference $A(3\Delta y_k) - A(\Delta y_k)$ by $A_0(3\Delta y_k) - A_0(\Delta y_k)$ in Eq. (21). The ratio $\varepsilon/\varepsilon_0 = \{[A_0(3\Delta y_k) - A_0(\Delta y_k)]/[A(3\Delta y_k) - A(\Delta y_k)]\}^{3/2}$ shows the degree of difference in the dissipation rate estimates with and without taking into account the averaging of the radial velocity over the sensing volume.

Figure 2 shows vertical profiles of $\Delta y_k$, $3\Delta y_k$, $F(\Delta y_k)$, $A(\Delta y_k)$, $A_0(\Delta y_k)$ $A(3\Delta y_k)$, $A_0(3\Delta y_k)$, $A(3\Delta y_k) - A(\Delta y_k)$, $A_0(3\Delta y_k) - A_0(\Delta y_k)$ and $\varepsilon/\varepsilon_0$. The dashed line corresponds to the value of the longitudinal dimension of the sensing volume calculated as $\Delta z = \Delta R/\mathrm{erf}\left(\Delta R/(2\Delta p)\right)$, where $\mathrm{erf}(x)$ is the error function (Banakh and Smalikho, 2013). It is seen that with increasing height $h_k$ the transverse dimension of the sensing volume increases and at heights greater than 400 m it becomes larger than the longitudinal dimension $\Delta z$, which does not depend on the measurement height. The $F(\Delta y_k)$ takes values of 5.8 m$^{2/3}$ at a height of 100 m and 8.3 m$^{2/3}$ at a height of 500 m (see Fig. 2 (b)). Fig. 2 (b) also illustrates the effect of spatial averaging of the radial velocity on the azimuth (transverse) structure function of the radial velocity within the inertial subrange of turbulence (if the condition $3\Delta y_k \ll L_V$ is satisfied). According to Fig. 2 (b), the ratio $A_0(\Delta y_k)/A(\Delta y_k)$ varies from 2.3 (at a height of 500 m) to 4.2 (at a height of 100 m) and the ratio $A_0(3\Delta y_k)/A(3\Delta y_k)$ varies from 1.4 (at a height of 500 m) to 2 (at a height of 100 m). As can be seen in Fig. 2 (c), the difference between $A(3\Delta y_k) - A(\Delta y_k)$ and $A_0(3\Delta y_k) - A_0(\Delta y_k)$ is much smaller and, according to Fig. 2 (d), the estimate of the dissipation rate without taking into account the averaging of the radial velocity over the sensing volume is understated by 1.5 times for a height of 100 m, and for heights above 375 m, the underestimation does not exceed 5%.

[revised manuscript text omitted]

The Stream Line lidar operated continuously during the experiment. The focus of the lidar beam was set to 500 m. The conical scanning with an angular rate of 6°/s (time of one full scan $T_{\mathrm{scan}} = 1$ min) at the elevation angle $\varphi = \varphi_E = 35.3°$ was used. The number of probing pulses for data accumulation was $N_a = 7500$, which corresponded to the duration of measurement for every azimuth scanning angle $T_a = 0.5$ s. In this case, for one full scan we have $M = T_{\mathrm{scan}} / T_a = 120$ such measurements with the resolution in the azimuth angle $\Delta\theta = 3°$. The range gate length $\Delta R$ was taken equal to 18m (vertical resolution $\Delta h = \Delta R \sin \varphi_E \approx 10$ m).

In the processing of data of these measurements, we set the minimum useful range $R_0 = 171$ m, which corresponded to a minimum 
[revised manuscript text omitted]
\}$, in accordance with Eq.(16), the estimate of the kinetic energy of turbulence can be represented as $E = (3/2)[\bar{\sigma}_{\mathrm{L}}^2 - \sigma_e^2 + \sigma_t^2]$, where the instrumental error in estimating the radial velocity $\sigma_e$ and the turbulent broadening of the Doppler spectrum $\sigma_t^2$ are determined using Eqs. (23) and (17), respectively. If $\sigma_e^2$ and $\sigma_t^2$ are negligible, in comparison with the variance of the lidar estimate of the radial velocity $\bar{\sigma}_{\mathrm{L}}^2$, an estimate of the kinetic energy with a sufficiently high accuracy can be obtained using the equation: $E = (3/2)\bar{\sigma}_{\mathrm{L}}^2$. To study the effect of $\sigma_e^2$ and $\sigma_t^2$ on the estimation of the kinetic energy, we obtained vertical profiles of $E_1 = (3/2)\bar{\sigma}_{\mathrm{L}}^2$ , $E_2 = (3/2)[\bar{\sigma}_{\mathrm{L}}^2 - \sigma_e^2]$ and $E_3 = (3/2)[\bar{\sigma}_{\mathrm{L}}^2 - \sigma_e^2 + \sigma_t^2]$. Four examples of such profiles are shown in Fig. 10. It can be seen that the allowance of the instrumental error $\sigma_e$ is important in the layer above 400 m, where the $\sigma_e$ increases due to a decrease in the signal-to-noise ratio SNR (see Figures 6 (d), 7 (b) and 8 (b)). A comparison of the red and blue curves in Fig. 10 allows one to judge the

effect of allowance of the spatial averaging of the radial velocity over the sensing volume on estimate of the turbulence kinetic energy. It follows from the data in Fig. 10 that the value $[(E_3 - E_2)/E_3] \times 100\%$ varies from 14% to 27% at a height of 100 m and from 10% to 16% at a height of 500 m. If for estimating the integral scale of turbulence $L_V$ in Eq. (24), instead of $E \equiv E_3$, to use $E_2$, then underestimation of the integral scale will be from 15% to 40%.

[revised manuscript text omitted]

The comparison of measurements of the turbulence energy dissipation rate by the Stream Line lidar with the method described in Section 3 and the data measured by the sonic anemometer has demonstrated a good agreement. The data of the lidar experiment of 2016 have been used to obtain the spatiotemporal distributions of different wind turbulence parameters with a height resolution of 10 m and a time resolution of 30 min. The lidar estimates of turbulence have been analyzed. It has been shown that the use of conical scanning during measurements by PCDL and the method for processing of lidar data proposed in this paper allows obtaining the information about wind turbulence in the atmospheric mixing layer with a rather high accuracy. However, as shown by the lidar experiment conducted under stable temperature stratification outside the layer of intensive turbulent mixing (Smalikho and Banakh, 2017), this method is not applicable and, consequently, further investigations and development of new approaches are needed.

**Appendix: List of symbols**

$B_\parallel(r)$            Longitudinal correlation function of wind velocity

$c$            Speed of light

$C_K \approx 2$            Kolmogorov constant

$C_1 = 8.4134$

$C_2 = 1.2717$

$C_3 = 0.0652$

$C_4 = 0.3796$

$D_r(\psi;\theta)$ — Azimuth structure function of the radial velocity

$D_r(\psi)$ — Azimuth structure function of the radial velocity for isotropic turbulence

$\bar{D}_r(\psi)$ — Averaged azimuth structure function of the radial velocity (Eq.(5))

$\bar{D}_L(\psi_l)$ — Azimuth structure function of lidar estimate of the radial velocity

$\bar{D}_a(\psi_l)$ — Azimuth structure function of the radial velocity averaged over the sensing volume

$D_\perp(y')$ — Transverse structure function of wind velocity

$E = (\sigma_w^2 + \sigma_u^2 + \sigma_v^2)/2$ — Kinetic energy of turbulence

$f_p$ — Pulse repetition frequency

$L_V$ — Integral scale of turbulence

$l_v$ — Inner scale of turbulence

$N$ — Number of conical scans

$N_a$ — Number of probing pulses used for the accumulation

$R$ — Range (distance from lidar)

$R_0$ — Minimum range

$R' = R\cos\varphi$ — Radius of the circle along which the sensing volume moves during the conical scanning

$r_H$ — Scale of the low-frequency boundary of the inertial subrange

$S_\|(\kappa)$ — Longitudinal spatial spectrum of wind velocity fluctuations

$S_\perp(\kappa)$ — Transverse spatial spectrum of wind velocity fluctuations

SNR — Signal-to-noise ratio

$T_{\mathrm{scan}}$ — Duration of one conical scan

$T_W$ — Temporal window width

$u$ — Fluctuations of longitudinal wind component

$U$ — Average wind velocity

$$ — Along-wind momentum flux

$v$ — Fluctuations of transverse wind component

$\mathbf{V} = \{V_z, V_x, V_y\}$ — Wind vector, where $V_z$ is the vertical component, $V_x$ and $V_y$ are the horizontal components

$V_a$ — Radial velocity averaged over the sensing volume

[revised manuscript text omitted]

**Responses for the reviewers of the manuscript**

We thank very much the reviewers for their time and efforts, thoughtful and very useful comments. We have incorporated
5 most of their suggested revisions as indicated below.

**Referee #1**
In this manuscript, the authors describe how various turbulent parameters can be measured with a continuously conically scanning Doppler lidar. The techniques for measurement of the parameters are described in detail, and sample results of the
10 measurements are shown. Doppler lidar measurements of the dissipation rate are compared with a sonic anemometer at 43 m, and are shown to generally agree well, except with some low biases under stable conditions when the lidar is unable to resolve the any portion of the inertial subrange. The turbulence kinetic energy from the Doppler lidar is shown to generally agree with measurements from a sonic anemometer at a lower height. In all, the scientific quality of the manuscript appears to be solidly based in theory and good. The work builds on previous work, with new refinements made to the strategy.
15 However, there are a few areas of the manuscript that could be clarified, as sections of the text are difficult to follow. As such, I recommend this manuscript be suitable for publication in AMT after minor revisions, in which the following comments, which are mostly of clarification, are addressed.
Specific Comments:
a) P. 1, line 19; p. 2, line 5 (and elsewhere): Change 'raw lidar data' to 'radial velocities'. By 'raw data', I interpret that to be
20 the measured Doppler spectrum, which are not used directly in the referenced techniques to measure turbulence.

The phrase "raw data measured" has been replaced by "measurements".

b) p. 2, line 9: By 'averaging over the sensing volume', clarify that you mean the spatial-temporal averaging of the pulse
25 length over one beam accumulation and not the averaging over the entire conical area.

Page 2, line 9: "(see Eq.(6) in paper of Smalikho and Banakh, 2013)" has been added.

c) p. 2 line 12: What are dr and σr?

Page 2, line 13: "and $\sigma_r^2 = B_{\parallel}(0)$ is the variance" and "radial" have been added.
"dr" is an infinitesimal increment of the integration variable "r" (separation between two points).

d) p. 2 line 20: Quantify 'high spatial resolution'.

Page 2, lines 21-22: "(longitudinal size of the sensing volume can be around 30 m)" has been added.

e) p. 2 line 23: What disadvantages of the earlier methods, precisely? The averaging over the sensing volume?

40 We do not know publications in which authors would take into account the effect of averaging of the radial velocity over the sensing volume when estimating the kinetic energy of turbulence.

f) p. 2 line 24: Change 'spatiotemporal' to 'time and height'. The term 'spatiotemporal' is too general, and generally means that information on the horizontal variability is measured/known.
45

Fixed.

g) p. 6 lines 22-24: This section is difficult to follow. Providing more text to describe the different terms and how they are related would be helpful.

Text on page 6 (lines 18-24) of initial version of the manuscript has been replaced by the text on page 6 (lines 19-26) and page 7 (lines 1-3) of the revised manuscript.

Page 7, line 13: "(Banakh and Smalikho, 2013)" has been added.

h) p. 7-8: For this section in particular, it would be helpful to add a figure providing a few examples of the 2-dimensional spectrum and showing how the different parameters are calculated from it (particularly interested in $\sigma_e$, $\sigma_t$), including adding a paragraph discussing the figures. This would be similar to showing how different parameters are calculated in Fig. 5.

Page 9, lines 3-13: The paragraph "With increasing range … without taking into account the averaging of the radial velocity over the sensing volume." has been added.

Page 9, lines 14-26: The paragraph "Figure 2 shows vertical profiles … the underestimation does not exceed 5%." has been added.

Page 24: Figure 2 has been added.

The sentence "The analysis of results for the kinetic energy of turbulence …  is understated by 10 - 20%, especially, in the layer up to 200 m." (page 13, lines 9-12 in the initial variant of the manuscript) has been removed.

Page 14, lines 22-30 and page 15, lines 1-4: The paragraph "Under the condition … then underestimation of the integral scale will be from 15% to 40%." has been added.

Page 32: Figure 10 has been added.

i) p. 10 line 10: How much of the data was unusable exactly? The percentage of unusable data would be helpful.

Page 11, line 16: "(around 15%)" has been added.

j) p. 10 line 13: What was the averaging time that the results shown in Fig. 3 were computed over? Based on p. 9 lines 19/24, it seems that 4 PPIs were used (over 5 minutes) while the sonic anemometer used 20 min of data. How were these differences in averaging times rectified?

If the same measurement time is used for the lidar and the sonic anemometer, the distance traveled by the sensing volume and the distance to which the air masses are carried by the mean wind during this time will vary greatly, since the velocity of the mean wind is substantially less than the linear velocity of movement of the sensing volume at the base of the scanning cone. We believe that in order to compare the results of estimating the dissipation rate, it is more appropriate to use the lidar data and the acoustic anemometer data, which correspond to the same distances.

k) p. 12 line 5: Is it possible to discern that the increase in kinetic energy computed over more scans (over longer time periods) is truly a better measure, and not simply due to non-stationarity of the mean wind (as discussed for the stable case at line 15) increasing the variances across the entire conical scan? Based on Fig. 6, the mean wind changes (wind speed slowly decreases, direction shifts) over the 6 hour time window mentioned, thus this may be causing the increase in measured TKE.

The variance of the average (30-minute averaging) of the wind velocity, calculated from the data in Figure 6 (a) for a height of 200 m and a time interval from 12:00 to 18:00, is approximately 10 times less than the TKE given in Table 1 (for 30 scans). Therefore, we can assume that the contribution of the nonstationarity of the mean wind to the kinetic energy estimate is negligible, in comparison with the turbulent fluctuations of the wind field. However, for another case considered in the manuscript (measurement at an altitude of 200 m from 01:00 to 07:00), the variance of the average (30-minute averaging) of the wind velocity is approximately twice the estimate of the kinetic energy obtained by using lidar data for 30 scans. This is the reason that, with an increase in the averaging interval from 10 min to 60 min, the magnitude of the kinetic energy estimate is monotonically increasing (it has no saturation, as in the first case under consideration). Apparently, for conditions of very weak turbulence on the background of nonstationarity of the mean wind, a special procedure for data filtering is required, which is not the subject of this paper.

l) p. 12 line 15: Other possible reasons include the inability of the lidar to resolve any portion of the inertial subrange (thus all derived parameters are not valid) and the low bias of dissipation (denominator for calculation of integral scale) when it is small.

We agree with this comment. Under conditions of stable thermal stratification of the atmosphere, the inertial subrange of turbulence can be much smaller than the size of the sensing volume, or even the inertial interval may be absent. It is obvious that the method of estimating the dissipation rate and the integral scale described in the manuscript is not applicable for this case. Therefore, in this manuscript there are no results of data processing, measured by the lidar in 2016 at night.

m) p. 12 line 20-22: The meaning and significance of 'The value of . . . over azimuth angles' is unclear; it should be rewritten.

Fixed.

n) p. 13 line 2: What is meant by 'close to each other'? A quantitative measure (standard deviation or range of values) is needed.

Page 14, lines 9-10: "(maximum deviation is around 20%)" has been added.

o) p. 15 line 125: Add the qualifier here that these high estimates were under stably stratified conditions.

Probably, the reviewer has in mind line 25.
Page 17, line 14: "(measurements in the daytime)" has been added.
Page 17, line 16: the sentence "Sometimes such estimates exceed 1 km in contrast to results shown in Figures 6(f), 7(d) and 8(d)." has been added.

p) End of manuscript: With the large number of variables and subscripts in this manuscript, adding a list of the symbols would be extremely helpful in reading this manuscript. I had to keep searching through the paper to find variables that were first introduced many pages earlier in the paper.

Pages 17-20: Appendix with a list of symbols has been added.

Technical corrections:
a) p. 6 line 10 (and reference list): 'Pearson' not 'Pierson'

Fixed.

b) p. 6 line 20: Should σe2 have an overbar as well?

Page 6, lines 25-26: The sentence "In Eqs. (13) and (14) it is assumed that $\sigma_e$ is independent of the azimuth angle $\theta_m$" has been added.

c) P. 11 line 15: 'continuously' is a better word than 'permanently'.

Fixed.

**Referee #2**
General comments:

This manuscript presents a methodology for deriving turbulent parameters from scanning Doppler lidar observations in the lower atmosphere. The methodology is sound and the results show that the parameters derived from Doppler lidar measurements usually agree well with reference parameters obtained from a sonic anemometer. The methodology uses a particular turbulence model which dictates how certain properties of the observed turbulence are expected to behave and so enable them to be derived.

A clear statement describing atmospheric situations when this model is applicable, and situations when it is not likely to be applicable, should be included in the conclusion. Are there methods for checking whether the turbulence model is applicable in a particular situation? For example, can you use the Doppler lidar observations to check for stationarity? In addition, what are the likely biases if the model is not strictly applicable, but provides reasonable results? An example here is the slight underestimates in turbulent energy dissipation rate provided by the Doppler lidar at low values. Is this expected because of unrealistic integral scales used, or is it an issue in accounting for radial velocity measurement uncertainty correctly?

To answer these questions, more research is needed. In this manuscript, we propose a method that is applicable for determining the parameters of wind turbulence from lidar measurements in the atmospheric layer of intensive mixing. The turbulence model, on the basis of which this method was developed, is quite applicable for such a layer. To obtain information about wind turbulence from measurements by a lidar in a stably stratified boundary layer (especially inside a low-level jet stream), it is necessary to apply another data processing procedure that is not known to us. Also it is necessary to take into account that at very strong stable temperature stratification the turbulence becomes intermittent and the inertial subrange can disappear.

Page 17, lines 23-25: The sentence "However, as shown by the lidar experiment conducted under stable temperature stratification outside the layer of intensive turbulent mixing (Smalikho and Bankh, 2017), this method is not applicable and, consequently, further investigations and development of new approaches are needed." has been added.

The manuscript contains all of the information necessary for publication, but in its current state is difficult to read. There are a huge number of variables and subscripts introduced, which although necessary for completeness, make it difficult to follow. It would be easier to comprehend if large parts of the derivation were placed in an appendix, with terms directly related to the parameters that will be derived from observations included in the text. In addition, the instrument should be introduced first in Section 3, so that it is easy to refer to the instrument specifications when introducing the measurement strategy. Add a table presenting the relevant instrument specifications, e.g. pulse-repetition-frequency, receiver bandwidth/Nyquist velocity, range gate length, azimuthal scanning speed, lidar wavelength, telescope type, rather than referring the reader to another paper. As an aid to the reader, this table could also include the associated variable in the equations. After some minor modifications, I feel this manuscript will be suitable for publication.

Pages 18-20: Appendix with a list of symbols has been added.
Main parameters of the Stream Line lidar are given in Table 1 of our paper published last year in AMT (see page 10, lines 9-10). In our opinion, the inclusion of this table in the manuscript submitted to the same journal would be superfluous. The parameters of the lidar experiments conducted in 2014 and 2016 differ and are given in Sections 4 and 5, respectively.

Specific comments:
Page 1, line 19: The data provided by these instruments is not really 'raw' data, but radial velocities.

The phrase "raw data measured" has been replaced by "measurements".

Page 2, line 13: Suggest replacing 'were proposed' by 'have been proposed'.

Fixed.

Page 2, line 24: Need to state that this is '100 to 500 m in altitude', as it could be assumed that the distances refer to range.

Fixed.

Page 2, line 27: Suggest starting the paragraph with 'First, we describe the equations that will be used to develop the measurement strategy and method for deriving the wind turbulence parameters:'

Page 2, lines 29, 30: "First of all, derive the equations to be used as a basis for development of the measurement strategy and the procedure of estimation of wind turbulence parameters:" has been replaced by "First, we describe the equations that will be used to develop the measurement strategy and the procedure of estimation of wind turbulence parameters:".

Page 2, line 28: The measured 'raw' radial velocities are not strictly instantaneous, as they are obtained by averaging a large number of samples internally.

Here we do not consider the radial velocity measured by a lidar.

Page 4, line 4: Suggest replacing 'some or other' with 'an appropriate'.

Fixed.

Page 4, lines 8-9: It would be clearer for the reader if these expressions were placed on separate lines.

Fixed.

Page 3, line 15; page4, lines 20-24; and Figure 1: It should be made clear, especially in the Figure caption, that the azimuth angle refers to the azimuthal resolution (if continuous scan) or separation between 2 adjacent rays in a scan (step-stare scan).

In Section 2 we find the condition under which the azimuth structure function of the radial velocity is equivalent to the spatial transverse structure function of the wind speed. Here we do not take into account the spatial averaging of the radial velocity over the sensing volume, which takes place in lidar measurements. For a transverse structure function, it is easy to take into account the spatial averaging over the sensing volume. In our experiments we used continuous scan and, therefore, the azimuth angle resolution is equal to the angle between two adjacent rays.

Page 5, line 1: Suggest replacing 'the both' with 'both'.

Fixed.

Page 5, line 5: What is the rationale behind choosing delta theta = 3 degrees? And what does L correspond to?

In principle, for calculation of the structure functions shown in Figure 1, we could choose any 'delta theta' which is less than 9 degrees (corresponding solid and dashed curves in Figure 1 almost coincide for azimuth angles less than 9 deg). In the case of 'delta theta' = 3 degrees and 'L' = 30 the maximum angle 'delta theta'*'L' = 90 degrees. The same 'delta theta' and 'L' were used to obtain structure functions shown in Figure 12 (in revised manuscript).

Page 5, Measurement strategy: Do you mean that you perform one conical scan with +ve azimuth rotation, then one scan with -ave azimuth rotation?

Yes.

Page 5, line 24: As defined previously, $R\_0$ should be (delta R / 2) if the first range gate is k=0, unless you define k=0 as the first usable range gate. Then 'minimal distance' should be defined precisely, e.g. define '$R\_0$ is the distance to the first usable range gate' before the equation on line 23, and explain why the first gate should satisfy the condition stated on line 25.

Page 5, lines 24 – 25: " $R_0$ is the distance to the first usable range gate" has been added.

"The minimal distance $R_0$ depends on the probing pulse duration. At the same time, it should satisfy the above condition $R_0 >> |< \mathbf{V} >|/(\omega_s \cos \varphi_E)$ ." has been removed. This condition must be satisfied for any ranges $R_k$ , as afore noted in Section 2 (see page 3, lines 16 – 17).

Page 5, line 26: The maximum range is effectively determined by the instrument pulse repetition frequency; the maximum usable range depends on the signal-to-noise ratio (SNR) and hence the atmosphere. Suggest rewriting this sentence, stating instead that the 'uncertainty in the radial velocity measurement depends on the SNR'.

Page 5, lines 26 – 27, page 6, lines 1 - 2: "The maximal distance … the true value of the velocity." has been replaced by "Uncertainty in the radial velocity measurement depends on the signal-to-noise ratio (SNR). At low SNR the probability of "bad" estimate …. To avoid the application of the data filtering procedure, … not contain "bad" estimates."

Page 6, line 9: Use correct reference (Pearson).

Fixed.

Page 6, line 11: Do you mean azimuthal dimension rather than longitudinal dimension here?

Page 6, line 12: "longitudinal" has been replaced by "transverse".

Page 6, line 14: How do you know if Lv only occasionally exceeds the sensing volume?

Page 6, line 15: "only few times exceeds the size of the sensing volume" has been replaced by "exceeds the size of the sensing volume insignificantly".

Page 6, lines 15-17: Other authors have shown that it is usually safer to always take account of the uncertainty in the radial velocity estimates.

Page 6, line 18: "(Frehlich et al., 2006)" has been added.

Page 6, lines 18-24: This sequence of equations requires much more explanation than is given here. ??

Text on page 6 (lines 18-24) of initial version of the manuscript has been replaced by the text on page 6 (lines 19-26) and page 7 (lines 1-3) of the revised manuscript.
Page 7, line 12: "(Banakh and Smalikho, 2013)" has been added.

Page 8, lines 12-15: Not sure that this can be justified without evidence.

Page 8, lines 18-23 and page 9, lines 1-2 (revised manuscript): The sentence "Since the instrumental error of estimation of the radial velocity … it is not necessary here to take into account the instrumental error and the effect from averaging of the radial velocity over the sensing volume." has been replaced by "Since the instrumental error of estimation of the radial velocity … to take into account the instrumental error and the effect from averaging of the radial velocity over the sensing volume. Indeed, as shown by Eberhard et al. (1989), in the case of a horizontally homogeneous turbulence statistics and …. Taking into account that …, Eq. (25) can also be regarded as exact.".

Page 9, line 16, and page 11, line 15: The focus of the lidar beam was set to XX m.

Fixed.

Page 11, line 9: Suggest 'To test the method for determining the kinetic energy,..'

5   Fixed.

Page 11, line 12-15: Suggest 'The presence of forest fires in the Tomsk region provided lidar measurements with high signal-to-noise ratios ...'

10  Fixed.

Page 11, line 15: Suggest replacing 'permanently' with 'continuously'.

Fixed.
15
Page 11, line 20: The 'minimum useful range'.

Fixed.

20  Page 12, line 3: I assume you mean 'horizontal wind speed'.

Page 13, line 11 (revised manuscript): "wind velocity" has been replaced by "horizontal wind speed".

Page 14, line 21: This assumes that the turbulent parameters don't change over the time required to obtain 30 scans.
25
Page 16, line 8 (revised manuscript): "In the case of stationary conditions" has been added.

Figure 3: Suggest replacing 'Time profiles of the turbulence' with 'Time series of the turbulent'.

30  Fixed.

Figure 4: Suggest replacing 'Time profiles' with 'Time series'.

Fixed.
35
Figure 6: Panel (a) should state 'Wind speed' rather than 'Wind velocity' for the colorbar title.

Usually in our publications in English we used "Wind velocity".

40  Figure 7: Suggest replacing 'Temporal profiles' with 'Time series'.

Fixed.

Figure 7,8: Suggest replacing 'instrumental error of estimation of the radial velocity' with 'uncertainty in radial velocity
45  estimate'.
Figure 9: Suggest replacing 'Spatiotemporal distributions' with 'Time-height plots', and 'relative error of estimation of the dissipation rate' with 'relative error in dissipation rate'.

Fixed.
50